# LEMUR: Learned Multi-Vector Retrieval

**Elias Jääsaari** [1]   **Ville Hyvönen** [1]   **Teemu Roos** [1]

## Abstract

Multi-vector representations generated by late interaction models, such as ColBERT, enable superior retrieval quality compared to single-vector representations in information retrieval applications. In multi-vector retrieval systems, both queries and documents are encoded using one embedding per token, and similarity between queries and documents is measured by the MaxSim similarity measure. However, the improved quality of multi-vector retrieval comes at the expense of significantly increased search latency. In this work, we introduce LEMUR, a simple yet efficient framework for multi-vector similarity search. LEMUR consists of two consecutive problem reductions: First, we formulate multi-vector similarity search as a supervised learning problem that can be solved using a one-hidden-layer neural network. Second, we reduce inference under this model to single-vector similarity search in its latent space, enabling the use of existing single-vector search indexes to accelerate retrieval. LEMUR is an order of magnitude faster than prior multi-vector similarity search methods. Our code is available at https://github.com/ejaasaari/lemur

## 1. Introduction

Embeddings generated by deep neural networks power modern information retrieval (IR) applications, including passage retrieval, document retrieval, and open-domain question answering. In the *single-vector* paradigm, each query and document is represented by a single vector in a shared vector space. The inner product between the query and document embeddings can be used to measure query-document similarity in an efficient fashion.

[1]Department of Computer Science, University of Helsinki, Helsinki, Finland. Correspondence to: Elias Jääsaari <elias.jaasaari@helsinki.fi>.

*Proceedings of the $43^{rd}$ International Conference on Machine Learning*, Seoul, South Korea. PMLR 306, 2026. Copyright 2026 by the author(s).

The latency of single-vector retrieval can be further reduced by leveraging approximate nearest neighbor search (ANNS) indexes and libraries, such as Faiss (Douze et al., 2026), HNSW (Malkov & Yashunin, 2018), DiskANN (Jayaram Subramanya et al., 2019), ScaNN (Guo et al., 2020), and LoRANN (Jääsaari et al., 2024). This enables scaling single-vector retrieval to massive document collections.

Khattab & Zaharia (2020) introduced ColBERT, whose late interaction mechanism enables *multi-vector* retrieval. Multi-vector models represent both queries and documents as sets of embeddings, with one embedding per token. In multi-vector retrieval, the similarity between a query and a document is measured by *MaxSim similarity*

$$\text{MaxSim}(X, C) = \sum_{x \in X} \max_{c \in C} \langle x, c \rangle,$$

where $X, C \subset \mathbb{R}^d$ denote the query and document embedding sets, respectively.

Due to their greater expressiveness enabled by fine-grained token-level representations, multi-vector models tend to have superior accuracy compared to single-vector models in IR applications (e.g., Khattab et al., 2021; Thakur et al., 2021; Lin et al., 2023). This has motivated the rapid development of new multi-vector models. After the introduction of ColBERT and ColBERTv2 (Santhanam et al., 2022b), better-performing (Jha et al., 2024; Amini et al., 2025; Clavié, 2025; Chaffin, 2025; Chaffin et al., 2026) and more memory-efficient (Takehi et al., 2025) multi-vector models have been introduced. In addition to retrieval from text corpora, multi-vector models such as ColPali (Faysse et al., 2025) have recently achieved state-of-the-art results in visual document retrieval (Günther et al., 2025; Xu et al., 2025; Teiletche et al., 2025; Moreira et al., 2026).

The improvement in the retrieval accuracy of multi-vector models comes at the cost of higher retrieval latency. This has motivated the development of multi-vector similarity search algorithms and systems, such as PLAID (Santhanam et al., 2022b), DESSERT (Engels et al., 2023), EMVB (Nardini et al., 2024), and IGP (Bian et al., 2025). These methods rely on *token-level pruning* of documents as the first step of their pipelines. For each query token, they first retrieve the most similar document tokens from the collection of all document tokens, and then restrict the search to documents containing the selected tokens. However, token-level

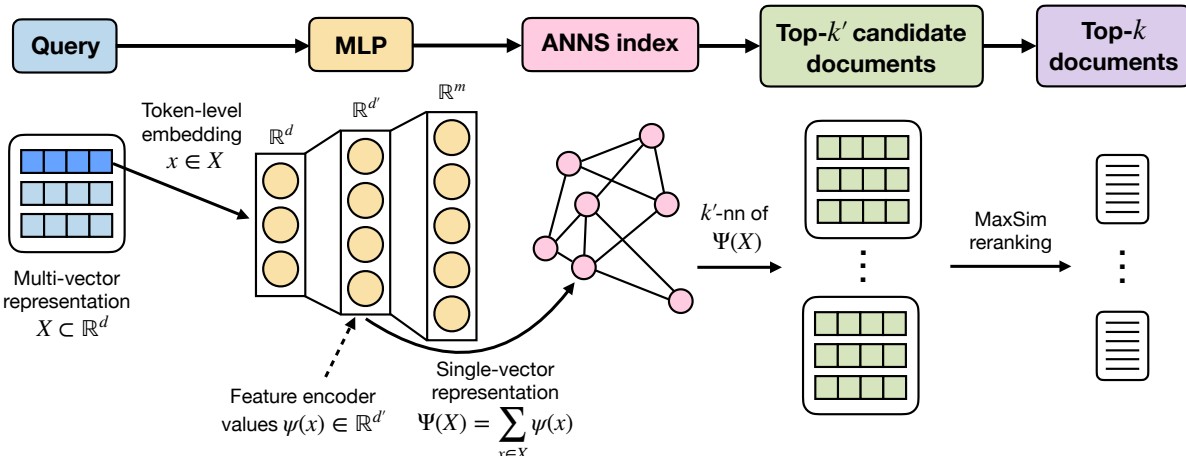

*Figure 1.* A schematic overview of the query-time process used by the LEMUR framework (for indexing, see Section 3). The latent representations $\psi(x)$ of the token-level embeddings $x \in X$ are obtained from the hidden layer of an MLP trained to estimate the MaxSim similarities between a query and each document. The single-vector representation $\Psi(X)$ is obtained by pooling these latent representations. The $k'$ documents most similar to $\Psi(X)$ are retrieved using an ANNS index. The final top-$k$ documents are selected by evaluating the exact MaxSim similarities to these $k'$ documents.

similarity between a query token and a document token is an inaccurate proxy for the MaxSim similarity of the corresponding query-document pair (Lee et al., 2023; Jayaram et al., 2024). Consequently, a large candidate set must be reranked to obtain accurate results.

MUVERA (Jayaram et al., 2024) does not rely on token-level pruning. Instead, it reduces multi-vector similarity search to single-vector similarity search by generating a single fixed-dimensional encoding (FDE) for each document and query. This single-vector reduction has enabled integrating MUVERA into widely used vector databases, including Weaviate and Qdrant. However, high-dimensional FDEs are required for accurate retrieval (see Figure 2), leading to large memory consumption and high latency.

In this paper, we introduce a simple yet effective framework (see Figure 1) for multi-vector similarity search. The framework is a fast corpus-specific search reduction that consists of two successive problem reductions: first to a supervised learning task and then to single-vector similarity search.

Our first key idea is that approximating the MaxSim similarity between a query and the corpus documents can be formulated as a supervised learning task. More specifically, it is a multi-output regression problem where there are as many outputs as there are documents in the corpus. This enables training a neural network directly for this task.

Our second key idea is that the MaxSim estimates of the model trained for this task are computed as inner products between two vectors that can be interpreted as single-vector representations of the query and the document. This makes it possible to reduce multi-vector similarity search to single-vector similarity search in the latent space of the model and

to leverage highly optimized single-vector ANNS libraries to speed up multi-vector retrieval on massive corpora. The design of LEMUR enables fast indexing and makes it applicable in both CPU- and GPU-based retrieval systems.

We call the proposed framework LEMUR (Learned Multi-Vector Retrieval). In addition to ColBERTv2 embeddings, we evaluate the performance of multi-vector similarity search methods on embeddings generated by modern multi-vector text models (Jha et al., 2024; Chaffin, 2025; Clavié, 2025; Amini et al., 2025) and visual document retrieval models (Faysse et al., 2025; Teiletche et al., 2025). LEMUR significantly outperforms the state-of-the-art methods on all datasets. The difference is especially pronounced on embeddings from models other than ColBERTv2.

In summary, our contributions are:

- We introduce LEMUR, a fast corpus-specific search reduction that reduces multi-vector similarity search over a corpus to single-vector similarity search by reformulating the problem as a supervised learning task.

- We evaluate the performance of multi-vector similarity search methods on 8 BEIR benchmark datasets (Thakur et al., 2021) embedded using 5 different multi-vector text models, and on the ViDoRe V3 visual retrieval dataset (Loison et al., 2026) embedded using 2 different multi-vector visual document retrieval models.

- We show that LEMUR outperforms the state-of-the-art multi-vector similarity search methods on all evaluated datasets, thus helping to bridge the latency gap between single-vector and multi-vector retrieval.

## 2. Background

### 2.1. Multi-Vector Retrieval

Multi-vector models represent both the query $\mathcal{X}$ and $m$ corpus documents $\{\mathcal{C}_j\}_{j=1}^m$ as sets of vectors by generating a $d$-dimensional contextualized embedding for each query and document token. Denote the query encoder and the document encoder of a multi-vector model by $Q$ and $D$, respectively. Further, denote the multi-vector representation of a query by $X = Q(\mathcal{X}) \subset \mathbb{R}^d$, and the multi-vector representations of the documents by $C_j = D(\mathcal{C}_j) \subset \mathbb{R}^d$ for each $j = 1, \ldots, m$. ColBERTv2 and most other multi-vector models truncate the query or pad it with [MASK] tokens so that all encoded queries have the same number of embeddings, whereas document representations have a variable number of embeddings.

In multi-vector retrieval, the similarity between the multi-vector representations of a query and a document is measured by *MaxSim* similarity (sometimes also referred to as the Chamfer similarity). The MaxSim similarity between a query $X$ and a document $C$ is defined as

$$\mathrm{MaxSim}(X, C) = \sum_{x \in X} \max_{c \in C} \langle x, c \rangle, \quad (1)$$

where each term is the similarity between a query token and the document token that is most similar to it, and the per-token similarity is measured by the inner product.

### 2.2. Single-Vector Similarity Search

In single-vector similarity search, also called approximate nearest neighbor search (ANNS), both the query and the documents are represented by single embedding vectors. Denote the single-vector representations of the query and the documents by $x \in \mathbb{R}^d$ and $\{c_j\}_{j=1}^m \subset \mathbb{R}^d$, respectively. The task is to identify the $k$ documents that are the most similar to the query, i.e., to approximate

$$\mathrm{NN}_k(x) := \{j \in [m] \,:\, s(x, c_j) \geq s(x, c^{(k)})\}, \quad (2)$$

where $c^{(1)}, \ldots, c^{(m)}$ denote the documents ordered with respect to their similarity to the query $x$ in descending order, and $s : \mathbb{R}^d \times \mathbb{R}^d \to \mathbb{R}$ is a similarity measure. When $s$ is the inner product, the corresponding problem is often called *maximum inner product search* (MIPS) (e.g., Guo et al., 2020; Lu et al., 2023; Zhao et al., 2023).

The quality of the approximation $S \subset [m]$, $|S| = k$, is typically measured by recall:

$$\mathrm{Recall}(S) = \frac{|S \cap \mathrm{NN}_k(x)|}{k}, \quad (3)$$

This is the fraction of the $k$ most similar documents correctly identified. Efficiency is typically measured by query latency or by queries per second (QPS) (Li et al., 2019; Aumüller et al., 2020; Jääsaari et al., 2025).

### 2.3. Multi-Vector Similarity Search

Compared to single-vector similarity measures, computing the MaxSim similarity is much more computationally expensive, since it requires evaluating the inner products between every query embedding and every document embedding. This makes *multi-vector similarity search* an even more performance-critical component in multi-vector retrieval. In multi-vector similarity search, the task is to approximate

$$\mathrm{NN}_k(X) := \{j \in [m] \,:\, s(X, C_j) \geq s(X, C^{(k)})\}, \quad (4)$$

where $C^{(1)}, \ldots, C^{(m)}$ denote the documents ordered with respect to their similarity to the query $X$ in descending order. This formulation is similar to the single-vector formulation (2), except now both the query $X \subset \mathbb{R}^d$ and the documents $C_j \subset \mathbb{R}^d$ are sets, and the similarity measure $s(\cdot, \cdot) = \mathrm{MaxSim}(\cdot, \cdot)$ operates on sets instead of vectors.

As in single-vector search, we measure the quality of the approximation $S \subset [m]$, $|S| = k$, by recall (3) with respect to the true MaxSim $k$-nearest neighbors, and we measure efficiency by query latency or by queries per second (QPS).

## 3. LEMUR Framework

In Section 3.1, we formulate multi-vector similarity search as a supervised learning task and outline how an MLP can be trained for this task. In Section 3.2, we then reduce multi-vector similarity search to single-vector similarity search in the learned latent space of the MLP.

### 3.1. Supervised Learning Formulation

Given a corpus represented by $\{C_j\}_{j=1}^m$, we consider the task of approximating the MaxSim similarities between a query $X$ and the documents. We use the approximate MaxSim similarities to select $k' > k$ candidate documents and then rerank them by exact MaxSim similarities.

Denote the target function by $f : 2^{\mathbb{R}^d} \to \mathbb{R}^m$, $f(X) = (f_1(X), \ldots, f_m(X))$, where

$$f_l(X) := \mathrm{MaxSim}(X, C_l)$$

for each $l \in [m]$. While the input of the target function $f$ is a variable-size set, the function can be written as

$$f(X) = \sum_{x \in X} g(x),$$

where $g : \mathbb{R}^d \to \mathbb{R}^m$ is defined by

$$g_l(x) := \max_{c \in C_l} \langle x, c \rangle$$

for each $l \in [m]$.[1] Thus, it suffices to estimate the function

---

[1]The function $g_l$ is commonly called the support function (Rockafellar, 1997) of $C_l$. The support function of a finite set of points is always a convex piecewise-linear function.

$g$ that operates on vectors and represents the contribution of a token-level embedding $x \in X$ to the sum (1).

Estimating $g$ is a multi-output regression problem with $m$ outputs, and an MLP $\phi : \mathbb{R}^d \to \mathbb{R}^m$ can be directly learned to solve this problem. We use the mean squared error (MSE) loss and train an MLP composed of a feature encoder $\psi : \mathbb{R}^d \to \mathbb{R}^{d'}$ and a linear output layer (without a bias) so that the function $g$ can be estimated as

$$g(x) \approx \phi(x) = W\psi(x),$$

where $W \in \mathbb{R}^{m \times d'}$. Because the output layer is linear, the final MaxSim estimates of the model can be computed as

$$f(X) \approx \sum_{x \in X} W\psi(x) = W \sum_{x \in X} \psi(x) = W\Psi(X), \quad (5)$$

where $\Psi(X) := \sum_{x \in X} \psi(x)$ denotes the pooled query feature vector.

### 3.2. Problem Reduction to Single-Vector ANNS

Since the number of documents $m$ can be large, the most computationally expensive part of the model inference is the computation of the matrix-vector product (5) at the output layer. However, note that we can write (5) as

$$f(X) \approx (\langle w_1, \Psi(X) \rangle, \dots, \langle w_m, \Psi(X) \rangle), \quad (6)$$

where the vectors $w_j \in \mathbb{R}^{d'}$, $j = 1, \dots, m$, are the rows of the weight matrix $W$. In other words, each of the $m$ outputs is given by the inner product of a query embedding and a document embedding, both represented in the same $d'$-dimensional latent space: the document embeddings $\{w_j\}_{j=1}^m$ are learned as the weights of the output layer, and the query embedding $\Psi(X)$ is obtained by pooling the feature encoder values $\psi(x)$.

Further, we do not need MaxSim estimates for all $m$ documents. Instead, we only need to find the $k'$ documents with the largest MaxSim estimates. Finding the $k'$ largest values in (6) is just a single-vector MIPS problem in the $d'$-dimensional latent space. Hence, given the pooled query embedding $\Psi(X)$, the documents with the largest MaxSim estimates can be retrieved using a single-vector ANNS algorithm with the weight vectors $\{w_j\}_{j=1}^m$ as the corpus.

Algorithm 1 summarizes the query-time pipeline in LEMUR.

Using the standard universal approximation theorem for MLPs (Cybenko, 1989; Hornik, 1991), it follows trivially (see Appendix B) that for every $\varepsilon > 0$, there exists a hidden dimension $d'$ such that $\|f(X) - W\Psi(X)\|_\infty \le |X|\varepsilon$ for any query $X$. Equivalently, for every document $l \in [m]$, $|\text{MaxSim}(X, C_l) - \langle w_l, \Psi(X) \rangle| \le |X|\varepsilon$.

---

**Algorithm 1** LEMUR Query Process

**Input:** Query embeddings $X$, feature encoder $\psi$, ANNS index built on $\{w_j\}_{j=1}^m$, document embeddings $\{C_j\}_{j=1}^m$, number of reranking candidates $k'$, output size $k$
**Output:** Top-$k$ documents for query $X$
1: $\Psi(X) \leftarrow \sum_{x \in X} \psi(x)$
2: $\mathcal{I} \leftarrow \text{ANN-TOPK}(\Psi(X), k')$
3: **for all** $j \in \mathcal{I}$ **do**
4:     $s_j \leftarrow \text{MaxSim}(X, C_j)$
5: **end for**
6: **return** the $k$ documents in $\mathcal{I}$ with the largest $\{s_j\}_{j \in \mathcal{I}}$

---

## 4. Implementation

In this section, we discuss how to implement the framework introduced in Section 3 in practice. First, we describe the network architecture we use. Then, we outline how model training can be scaled to large corpora by pretraining the feature encoder on a subset of the documents. Finally, we show that LEMUR is robust to the choice of training data.

### 4.1. Network Architecture

We use a small two-layer MLP as the network $\phi$. This allows us to keep the inference cost trivial and to train the network quickly, even on a CPU. The feature encoder $\psi$ is defined as

$$\psi(x) = \text{LN}(\text{GELU}(W'x + b)),$$

where $\text{LN}(\cdot)$ is layer normalization (Ba et al., 2016) and $\text{GELU}(\cdot)$ is the GELU activation function (Hendrycks & Gimpel, 2016), $W' \in \mathbb{R}^{d' \times d}$, and $b \in \mathbb{R}^{d'}$.

We did not observe significant performance gains from increasing the network depth beyond two layers, since each $g_l$ is a convex piecewise-linear function. Based on our ablation study (see Section 6.2), we fix the hidden layer size to $d' = 2048$ in all other experiments. Increasing $d'$ further improves the accuracy of the MaxSim estimates but has only a small impact on the end-to-end performance since it also increases the computational cost of retrieval.

In all experiments, we always use the same hyperparameters to train the model; see Appendix A.

### 4.2. Scalable Model Training

To scale model training to large corpora, we pretrain the feature encoder $\psi$ using a subset of the documents as targets. Specifically, we sample $m' \ll m$ documents and define the target function $g' : \mathbb{R}^d \to \mathbb{R}^{m'}$, $g' = g_{I'}$, where $I' \subset [m]$ is the set of indices of the $m'$ sampled documents. Using these auxiliary targets, we train a model $\phi' : \mathbb{R}^d \to \mathbb{R}^{m'}$,

$$\phi'(x) = W''\psi(x),$$

where $W'' \in \mathbb{R}^{m' \times d'}$.

We fix the weights of the feature encoder $\psi$ and learn the $j$th row of the output layer's final weight matrix $W$ for the final model $\phi$ by solving

$$w_j := \underset{\beta \in \mathbb{R}^{d'}}{\arg\min} \; \mathbb{E}\left[\left(\beta^\top \psi(x) - g_j(x)\right)^2\right] \qquad (7)$$

for each $j \in [m]$.

When the weights of $\psi$ are fixed, (7) is a standard linear regression task with a closed-form OLS solution. To speed up computation, we sample a smaller training set of size $n'$ to compute the OLS solutions.

After computing the OLS solutions for all $m$ documents, the rows $w_l$ of the weight matrix $W$ can be stored in an ANNS index. Although we present CPU-only results in this paper, we note that both the indexing and query processes of LEMUR are also easy to implement on a GPU, using, for example, CAGRA (Ootomo et al., 2024) for GPU-based ANNS. Algorithm 2 summarizes the indexing pipeline.

**Indexing speed and index updates.** In our experimental setup (see Section 6.1), on our largest dataset (8.8 million documents), training the feature encoder $\psi$ takes 20 minutes on a CPU. Computing the weight matrix $W$ of the final projection layer takes 70 minutes, and building the ANN graph takes 198 minutes, for a total of 4.8 hours. The time to compute $W$ scales linearly with the number of documents, while ANN index construction is often near $O(n \log n)$ for graph indexes. We note that we have not tuned our hyperparameters (see Appendix A) with respect to index construction time and that indexing can be sped up significantly with only a minor decrease in performance.

Compared to prior methods, LEMUR is more scalable to large datasets because it does not require clustering all token embeddings (e.g., 600 million embeddings into $2^{18}$ clusters on our biggest dataset), a step that is infeasible without a GPU on large datasets. Moreover, LEMUR indexes much lower-dimensional vectors than MUVERA (see Section 6.2). Since the feature encoder is kept fixed, new documents can be indexed quickly in LEMUR by computing the document weights via linear regression and inserting them into the ANNS index (with graph-based ANNS methods such as HNSW supporting online updates).

### 4.3. Training Set Selection

Because only some of the datasets used in our experiments have a separate set of training queries, we use the corpus documents encoded with the query encoder to train LEMUR in all experiments for consistency. Formally, we sample a small subset of $n' \ll m$ corpus documents $\mathcal{C}_1, \ldots \mathcal{C}_{n'}$. We encode them with the query encoder of the multi-vector model to obtain $X_i = Q(\mathcal{C}_i)$ for each $i \in [n']$, and sample $n$ token embeddings for the training set from the set $\bigcup_{i=1}^{n'} X_i$.

---

**Algorithm 2** LEMUR Indexing Process

**Input:** Corpus embedding sets $\{C_j\}_{j=1}^m$, training token embeddings $\{x_i\}_{i=1}^n$, number of auxiliary targets $m'$, OLS sample size $n'$

**Output:** Feature encoder $\psi$, ANNS index over $\{w_j\}_{j=1}^m$

1: Sample $I' \subset [m]$ with $|I'| = m'$
2: Define the auxiliary target function $g' = g_{I'}$
3: Train MLP $\phi'(x) = W'' \psi(x)$ to estimate $g'(x)$
4: $Z \leftarrow [\psi(x_1), \ldots, \psi(x_{n'})]^\top \in \mathbb{R}^{n' \times d'}$
5: **for all** $j \in [m]$ **do**
6: $\quad y_j \leftarrow [g_j(x_1), \ldots, g_j(x_{n'})]^\top \in \mathbb{R}^{n'}$
7: $\quad w_j \leftarrow Z^+ y_j$
8: **end for**
9: $\mathcal{J} \leftarrow \text{BUILD-ANNS-INDEX}(\{w_j\}_{j=1}^m)$
10: **return** $\psi$ and $\mathcal{J}$

---

If there is a separate training set of queries $\mathcal{X}_1, \ldots, \mathcal{X}_{n'}$ available, the query encoder of the multi-vector model can be used to generate the multi-vector representations $X_1, \ldots, X_{n'}$, where $X_i = Q(\mathcal{X}_i)$, and the training set can be sampled from $\bigcup_{i=1}^{n'} X_i$. Our results (see Appendix H.2) indicate that using actual queries (when available) for training yields even higher performance.

Notably, even when the training set is sampled directly from the embeddings of the corpus $C_1, \ldots, C_m$ (encoded using the document encoder $D$), LEMUR achieves consistent performance and still outperforms the baseline methods (see Appendix H.1). This training method requires neither additional data nor the use of an encoder. In summary, our results suggest that LEMUR is robust with respect to the type of data used to train the model.

**Random hidden layer weights.** We hypothesize that the robustness of LEMUR is because both query and document tokens are mapped by design by the late interaction model into a common semantic embedding space. As a result, although the training sets differ, they all provide samples from a similar underlying geometry.

In Appendix H.3, we perform an experiment in which we do not train the feature encoder $\psi$ at all; instead, we use an MLP with *random weights* in the hidden layer. Thus, the only component that adapts to the training data is the final projection layer. This is often referred to as an extreme learning machine (ELM) (Huang et al., 2006). Although performance degrades relative to training the feature encoder, LEMUR still works well in this scenario, suggesting that its performance does not primarily depend on learning a representation highly specific to the training data. Instead, the hidden layer mainly serves to expand the input into a richer feature space. Different training sets may slightly change the distribution of examples, but do not change the underlying geometry to which the linear layer is fitted.

# 5. Related Work

## 5.1. Multi-Vector Similarity Search

The retrieval system of ColBERT (Khattab & Zaharia, 2020) relies on *token-level pruning* of documents: For each query token, the most similar document tokens are first retrieved from $\bigcup_{j=1}^{m} C_j$, i.e., from the set of all document tokens. The documents containing the retrieved tokens are reranked by computing their MaxSim similarities to the query. PLAID (Santhanam et al., 2022a), DESSERT (Engels et al., 2023), EMVB (Nardini et al., 2024), and IGP (Bian et al., 2025) improve latency and memory efficiency of this basic scheme by adding different pruning and approximation steps. WARP (Scheerer et al., 2025) is an efficient retrieval engine for models trained specifically with the XTR (Lee et al., 2023) objective.

Like LEMUR, MUVERA (Jayaram et al., 2024) reduces multi-vector similarity search to single-vector similarity search by generating single-vector encodings for queries and documents, after which any single-vector ANNS library can be used for retrieval. However, the representations of MUVERA are data-oblivious, whereas the representations of LEMUR are learned in a supervised fashion by a model that is directly optimized for the task of interest. This makes LEMUR representations more accurate at much lower dimensionality (see Section 6.2) and more robust across different embedding models (see Section 6.3).

## 5.2. Distilling MaxSim Scores

TCT-ColBERT (Lin et al., 2020; 2021) learns a single-vector dense retriever by distilling ColBERT's MaxSim scores. BGE-M3 (Chen et al., 2024) also uses self-knowledge distillation between dense, sparse, and ColBERT-style multi-vector retrieval scores, but trains a unified embedding model rather than a corpus-specific search reduction. In contrast, LEMUR does not train a generic text encoder for queries and documents: it learns a corpus-specific search reduction for an existing multi-vector model. This makes LEMUR a lightweight, task-agnostic search approximation that avoids expensive dense-retriever training and can be applied directly to existing late interaction models.

## 5.3. Reducing the Storage Cost of Multi-Vector Models

Compared to single-vector retrieval, multi-vector retrieval methods have both larger memory footprints and higher latencies because of the large number of embeddings that have to be stored and processed. There exist many recent works (e.g., Clavié et al., 2024; MacAvaney et al., 2025; Yan et al., 2025; Veneroso et al., 2025; Qin et al., 2026) that address the increased storage cost by decreasing the number of embeddings per document. These methods are orthogonal to our approach and can be combined with it.

*Table 1.* Datasets used in the experiments.

| dataset | # corpus | # queries | # embeddings (average) |
|---|---|---|---|
| MSMARCO | 8 841 823 | 6 980 | 67.5 |
| HotpotQA | 5 233 329 | 7 405 | 58.6 |
| NQ | 2 681 468 | 3 452 | 85.2 |
| Quora | 522 931 | 10 000 | 15.6 |
| Touche-2020 | 382 545 | 49 | 104.6 |
| TREC-Covid | 171 332 | 50 | 118.6 |
| FiQA-2018 | 57 638 | 648 | 105.0 |
| SCIDOCS | 25 657 | 1 000 | 147.0 |
| ViDoRe | 12 969 | 100 | 1073.3 |

# 6. Experiments

## 6.1. Experimental Setup

Each experiment is run on a compute node with two 20-core Intel Xeon Gold 6230 (Cascade Lake) CPUs. Each CPU supports AVX-512 instructions, and we run each experiment on all 40 cores by parallelizing over queries.

**Datasets.** For text models, we use eight retrieval datasets from the BEIR (Thakur et al., 2021) benchmark. Following Santhanam et al. (2022a) and Jayaram et al. (2024), we use the development set for MS MARCO and the test set for all other datasets. For visual document retrieval models, we pool together images and queries from the five English-language ViDoRe V3 (Loison et al., 2026) datasets. Table 1 lists the number of documents, the number of test queries, and the average number of embeddings per document for each dataset (using ColBERTv2 for the text datasets and ColQwen2 for ViDoRe). We use the value $k = 100$ in all experiments in this section; for additional results with $k = 10$, see Appendix F.2.

**Embedding models.** In our main experiments, we use the widely used ColBERTv2 (Santhanam et al., 2022b) model. In Section 6.3, we also measure end-to-end performance on the answerai-colbert-small-v1 (Clavié, 2025), GTE-ModernColBERT-v1 (Chaffin, 2025), LFM2-ColBERT-350M (Amini et al., 2025), and jina-colbert-v2 (Jha et al., 2024) text retrieval models, and on the visual document retrieval models ColModernVBERT (Teiletche et al., 2025) and ColQwen2-v1.0 (Faysse et al., 2025).

## 6.2. Ablation Study

**Latent space dimension.** By varying the hidden layer size $d'$ of the model $\phi$, we study the effect of the dimensionality of the single-vector embeddings generated by LEMUR on retrieval accuracy. Specifically, we measure the proportion of the true MaxSim 100-nn of a query contained in the top-$k'$ candidates (Recall100@$k'$) selected by exact infer-

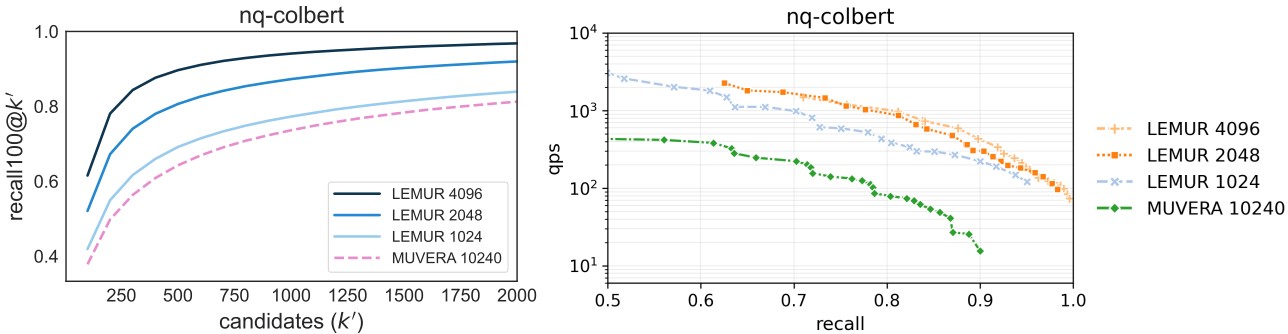

*Figure 2.* Ablation study on the effect of the hidden layer size $d'$ on the performance of LEMUR. Left: Comparison of Recall100@$k'$ for three values of $d'$ as a function of the candidate set size $k'$. Right: Comparison of the end-to-end performance between different values of $d'$ with ANNS and reranking included. While larger values of $d'$ can yield more accurate estimates, the end-to-end performance gap is narrower due to increased ANNS complexity, yielding diminishing returns.

ence with the model $\phi$. Increasing the dimension of the latent space consistently improves recall (see Figure 2, left). As a baseline, we use 10240-dimensional FDEs generated by MUVERA. The results indicate that the single-vector embeddings generated by LEMUR are significantly more accurate than FDEs: on seven out of eight datasets (see Appendix F), even the smallest 1024-dimensional embeddings yield a higher recall than the $10\times$ larger FDEs.

We then study the effect of the latent-space dimension $d'$ of the model on end-to-end latency (see Figure 2, right, and Appendix F). While increasing $d'$ from 1024 to 2048 significantly reduces latency, there is a point of diminishing returns when $d'$ is increased to 4096: even though a larger latent space dimension yields higher-quality candidates (as established in Figure 2, left), it also increases the computational cost of querying the ANNS index, which partially offsets the accuracy gains. This also explains why the performance gap to 10240-dimensional FDEs is even larger in the end-to-end setting. While using the value $d' = 4096$ leads to lower latency than $d' = 2048$ on some of the datasets (see Appendix F), the differences in performance are small. Hence, we use $d' = 2048$ in our end-to-end experiments to reduce memory consumption.

**MaxSim correlation.** In Appendix D, we measure Pearson's correlation coefficient and Spearman's rank correlation coefficient between LEMUR's estimates and the exact MaxSim similarities. Both correlation coefficients are high ($> 0.94$) for each dataset.

### 6.3. End-to-End Performance

In this section, we compare the end-to-end performance of LEMUR against MUVERA (Jayaram et al., 2024), DESSERT (Engels et al., 2023), IGP (Bian et al., 2025), and PLAID (Santhanam et al., 2022a). We use recall to measure effectiveness, and queries per second (QPS) to measure the efficiency of the algorithms.

**LEMUR implementation.** For LEMUR, we use PyTorch for both training and inference for the model described in Section 4. As a single-vector similarity search library, we use Glass (Wang, 2025), an efficient implementation of HNSW (Malkov & Yashunin, 2018) with scalar quantization. We implement MaxSim reranking in C++.

**Baseline implementations.** For MUVERA, we use the official C++ implementation to generate the FDEs. To ensure a fair comparison, we also combine MUVERA's FDEs with Glass and the same reranking implementation used for LEMUR. For DESSERT and IGP, we use the official C++ implementations, and for PLAID, we use the Rust implementation in the fast-plaid library (Sourty, 2025).

**Index hyperparameters.** For both LEMUR and the baseline methods, we fix the index hyperparameters to values recommended by the authors, except for DESSERT, for which we perform a grid search over $L \in \{32, 64\}, C \in \{5, 7\}$. Specifically, for LEMUR, we set the hidden layer size to $d' = 2048$; for MUVERA, as recommended by Jayaram et al. (2024), we set $R_{\text{reps}} = 40$, $k_{\text{sim}} = 6$, $d_{\text{proj}} = d$ and then apply a final projection to generate 10240-dimensional FDEs; for PLAID, DESSERT, and IGP, we set the number

*Table 2.* Best QPS at $\geq 80\%$ recall ($k = 100$). A dash indicates that the method did not reach the specified recall.

| Dataset | lemur | muvera | igp | dessert | plaid |
|---|---|---|---|---|---|
| MSMARCO | **799** | 150 | 62 | – | 13 |
| HotpotQA | **426** | 22 | 37 | – | 10 |
| NQ | **869** | 79 | 107 | 38 | 16 |
| Quora | **4068** | 787 | 679 | 284 | 89 |
| Touche-2020 | **1996** | 371 | 47 | 147 | 29 |
| TREC-Covid | **1753** | 57 | 34 | 109 | 23 |
| FiQA-2018 | **2416** | 239 | 310 | 242 | 51 |
| SCIDOCS | **2591** | 391 | 320 | 285 | 85 |

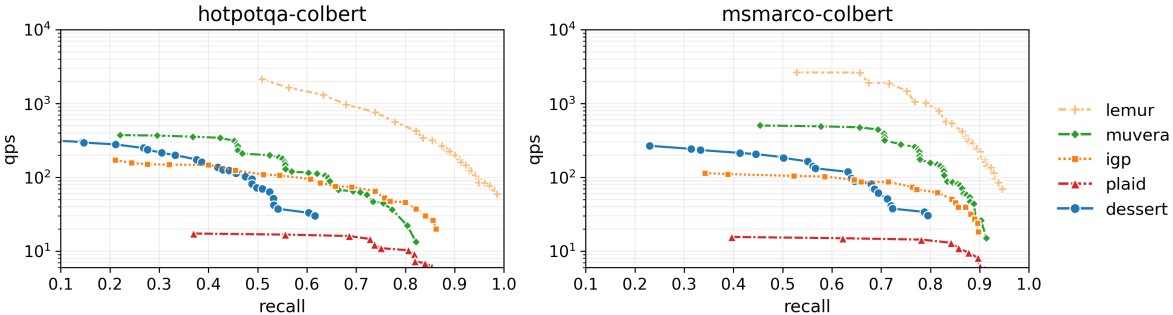

*Figure 3.* End-to-end performance comparison using ColBERTv2 embeddings on the HotpotQA (left) and MS MARCO (right) datasets. On both datasets, LEMUR is significantly faster than the baseline methods.

of clusters to $16\sqrt{n}$ (rounded down to the nearest power of two), where $n$ is the number of corpus token embeddings.

**Query hyperparameters.** For LEMUR and MUVERA, we vary the candidate-list-size parameter (`ef_search`) of the ANNS library and the number of reranked documents ($k'$). For the other methods, we vary the number of clusters probed and the number of reranked documents. We perform a grid search over the query hyperparameters and report results for the Pareto-optimal hyperparameter combinations.

**ColBERTv2 embeddings.** First, we consider BEIR datasets embedded using the ColBERTv2 model (see Table 2, Figure 3, and Appendix G.1). LEMUR consistently outperforms the baseline methods. In particular, at $\geq 80\%$ recall, it is 5–16× faster than the best-performing baseline method.

In Appendix C, we use nDCG@10 on the BEIR datasets to verify that LEMUR also preserves the retrieval quality of ColBERTv2 while maintaining low latency.

**Other text embedding models.** We then consider the BEIR datasets Quora (see Figure 4) and SCIDOCS (see Appendix G.2) embedded using the answerai-colbert-small-v1, GTE-ModernColBERT-v1, LFM2-ColBERT-350M, and jina-colbert-v2 models. Again, LEMUR significantly outperforms the baseline methods. We also observe that some of the baseline methods struggle with these embeddings. In particular, MUVERA fails to exceed 60% recall on the answerai-colbert-small-v1, GTE-ModernColBERT, and LFM2-ColBERT embeddings.

**Visual document models.** Finally, we consider the ViDoRe dataset embedded using the ColModernVBERT and ColQwen2-v1.0 models (see Figure 5). LEMUR outperforms the baseline methods in both cases. The performance gap is smaller than on the text datasets due to the proportionally much higher cost of the reranking stage caused by bigger document (image) representations. Since our default training set selection strategy is not available for visual document retrieval models, we simply use documents embedded using the document encoder $D$ as a training set (the third strategy mentioned in Section 4.3); using a separate query set for

training yields even better performance (see Appendix H.2).

**Long queries.** While the default number of query vectors used in ColBERTv2 is 32, for long queries, e.g., in agentic applications, using more vectors can be preferable. To test the effectiveness of LEMUR with longer queries, in Appendix E, we use the ArguAna dataset from BEIR with 300 vectors per query, following Santhanam et al. (2022b). LEMUR still maintains both retrieval performance and MaxSim approximation quality.

## 7. Discussion

Because of their fine-grained representations that capture token-level interactions, multi-vector models are more expressive and, consequently, tend to have higher retrieval quality than single-vector models. However, the high latency of multi-vector retrieval engines has hindered their wider adoption. To address this issue, we introduce LEMUR, a simple yet efficient framework that is an order of magnitude faster than earlier multi-vector similarity search methods, enabling efficient multi-vector retrieval at scale. Moreover, LEMUR produces single-vector representations for queries and documents, reducing the task to single-vector similarity search. This makes it easy to integrate LEMUR with state-of-the-art single-vector ANNS libraries.

Because they are learned in a supervised fashion, the single-vector embeddings generated by LEMUR are significantly more accurate than the fixed-dimensional encodings (FDEs) (Jayaram et al., 2024) that have been used for single-vector reduction earlier: in particular, 1024-dimensional LEMUR embeddings yield higher recall than 10240-dimensional FDEs (see Section 6.2). This enables using lower-dimensional single-vector embeddings, which reduces the memory footprint and leads to lower end-to-end latency due to faster single-vector similarity search. Standard scalar- or product-quantization (Jégou et al., 2011) techniques can be applied to the single-vector embeddings generated by LEMUR to further reduce memory consumption and end-to-end latency.

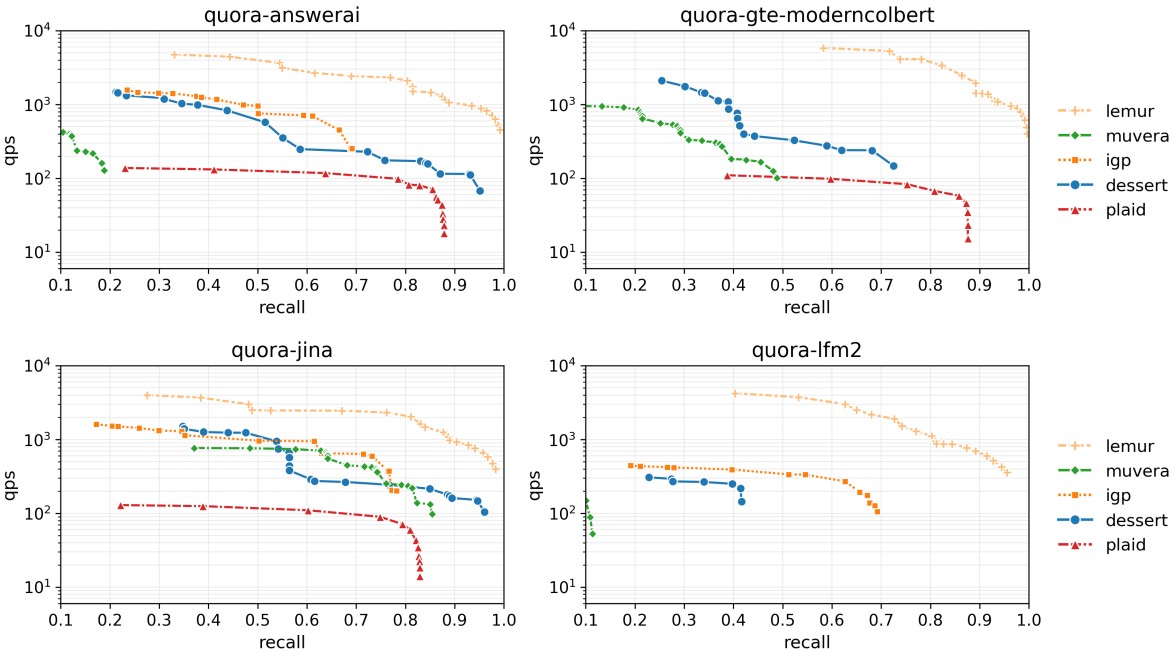

*Figure 4.* End-to-end performance comparison using four modern multi-vector text models on the Quora dataset. Across all models, LEMUR is significantly faster than the baseline methods, with MUVERA struggling especially on the non-ColBERTv2 models.

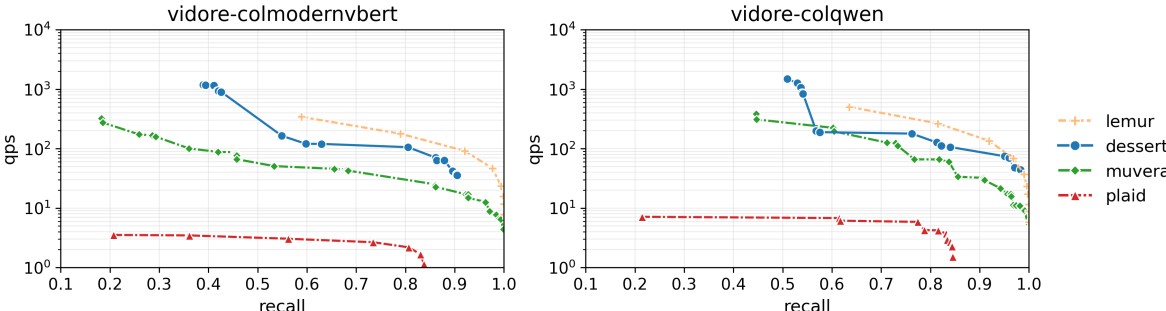

*Figure 5.* End-to-end performance comparison using two visual document retrieval models on the ViDoRe dataset. LEMUR yields state-of-the-art performance compared to the baselines, but the gap is narrower than for the text models.

The performance of earlier multi-vector similarity search methods has been evaluated almost exclusively on Col-BERTv2 (Santhanam et al., 2022b) embeddings. We conduct a performance evaluation on four additional multi-vector text models and two multi-vector visual document retrieval models (see Section 6.3). Our results indicate that the model used to generate the embeddings has a larger impact on the performance of multi-vector similarity search methods than the embedded dataset. In particular, most earlier methods either fail to achieve acceptable recall levels at all or have prohibitively high latency on the LFM2-ColBERT-350M and answerai-colbert-small-v1 embeddings. In contrast, LEMUR consistently achieves the highest recall levels with low latency regardless of the multi-vector model used to generate the embeddings. These observations underline the importance of establishing robust benchmarking practices for multi-vector similarity search.

**Limitations and future work.** While we do not optimize for memory consumption of the multi-vector representations in this work, LEMUR can be combined with e.g. product quantization (Jégou et al., 2011) or scalar quantization. We leave compatibility with extremely low-precision compression, such as 2-bit quantization, as future work.

While the MLP in LEMUR can be trained quickly and without a separate training set, even using the corpus itself as a training set may still pose an operational challenge when building an index incrementally as documents arrive. Our experiments show that using random hidden-layer weights is feasible (see Section 4.3), but we leave the investigation of synthetic and cross-corpus training sets as future work.

Finally, future work will examine whether the benefits of framing MaxSim as multi-output regression generalize to other similarity measures and retrieval tasks.

## Impact Statement

This work improves the efficiency of multi-vector retrieval and is foundational machine learning research, and thus does not have a direct societal impact.

## Acknowledgments

This work has been supported by the Research Council of Finland (grant #361902 and the Flagship programme: Finnish Center for Artificial Intelligence FCAI). The authors acknowledge the research environment provided by ELLIS Institute Finland, as well as CSC – IT Center for Science, Finland, for computational resources.

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

## A. LEMUR Hyperparameters

In this section, we list the hyperparameters used to train the LEMUR model (see Section 4). LEMUR is robust to the hyperparameters, and we use the same hyperparameters for all experiments in this paper. We did not observe benefits from using regularization because the model is a shallow network learning an exact, noise-free function.

*Table 3.* LEMUR model training hyperparameters.

| | |
|---|---|
| Corpus points sampled as outputs for model training ($m'$) | 8192 |
| Token embeddings sampled as model training set ($n$) | 100 000 |
| Token embeddings sampled for computing the OLS solutions ($n'$) | 16 384 |
| Optimizer | Adam (Kingma & Ba, 2015) |
| Learning rate | 0.003 |
| Epochs | 100 |
| Batch size | 512 |
| Gradient clip | 0.5 |

We also standardize the outputs for training the MLP using the global mean and standard deviation of the output values.

We use the Glass (Wang, 2025) library for ANNS with graph degree $R = 64$ and insertion-time beam width $L = 800$.

## B. LEMUR Universal Approximation

We assume that the MLP used in LEMUR is universal on compact sets. Concretely, for any compact set $K \subset \mathbb{R}^d$, any continuous $h : K \to \mathbb{R}^m$, and any $\varepsilon > 0$, there exists a hidden dimension $d'$ such that

$$\sup_{x \in K} \|h(x) - W\psi(x)\|_\infty \leq \varepsilon.$$

This is the standard universal approximation property for MLPs on compact domains (Cybenko, 1989; Hornik, 1991).

**Theorem B.1.** *Let $K \subset \mathbb{R}^d$ be compact, and suppose every token-level query embedding lies in $K$. Then, for every $\varepsilon > 0$, there exist a hidden dimension $d'$, a feature map $\psi : \mathbb{R}^d \to \mathbb{R}^{d'}$, and a matrix $W \in \mathbb{R}^{m \times d'}$ such that for all $x \in K$, $\|g(x) - W\psi(x)\|_\infty \leq \varepsilon$. Consequently, for every query $X \subset K$, $\|f(X) - W\Psi(X)\|_\infty \leq |X| \varepsilon$. Equivalently,*

$$\left| \text{MaxSim}(X, C_l) - \langle w_l, \Psi(X) \rangle \right| \leq |X| \varepsilon$$

*for every document $l \in [m]$,*

*Proof.* For each fixed $l \in [m]$, the function $g_l$ is the maximum of finitely many linear functions, and hence it is continuous on $\mathbb{R}^d$. Therefore, $g : K \to \mathbb{R}^m$ is continuous on $K$.

By the universal approximation property, for any $\varepsilon > 0$ there exist $d'$, $\psi$, and $W$ such that

$$\sup_{x \in K} \|g(x) - W\psi(x)\|_\infty \leq \varepsilon.$$

Now fix any query $X \subset K$. We have

$$f(X) - W\Psi(X) = \sum_{x \in X} \big( g(x) - W\psi(x) \big).$$

Taking the $\ell_\infty$ norm and applying the triangle inequality yields

$$\|f(X) - W\Psi(X)\|_\infty \leq \sum_{x \in X} \|g(x) - W\psi(x)\|_\infty \leq \sum_{x \in X} \varepsilon = |X| \varepsilon.$$

Taking the $l$th coordinate gives

$$\left| \text{MaxSim}(X, C_l) - \langle w_l, \Psi(X) \rangle \right| \leq |X| \varepsilon.$$

$\square$

## C. Retrieval Quality Evaluation

In this section, we use nDCG@10 (Järvelin & Kekäläinen, 2002) to verify that LEMUR preserves the retrieval quality of ColBERTv2 in the experiments of Section 6.3 and Appendix G.1. For each dataset, we list in the table below the nDCG@10 obtained using exact MaxSim search and then list the best QPS achieved by LEMUR such that the retrieved results match the nDCG@10 obtained using exact search within the specified $\varepsilon$. LEMUR is able to match the retrieval quality of ColBERTv2 while maintaining high QPS.

|  | SCIDOCS | FiQA | TREC-COVID | Touche | Quora | NQ | HotpotQA |
|---|---|---|---|---|---|---|---|
| nDCG@10 | 0.158 | 0.347 | 0.727 | 0.257 | 0.857 | 0.561 | 0.678 |
| QPS ($\varepsilon < 0.01$) | 3142 | 2753 | 1994 | 2389 | 4068 | 917 | 108 |
| QPS ($\varepsilon < 0.001$) | 1351 | 1038 | 708 | 2389 | 2191 | 179 | 77 |

## D. Correlation with Exact MaxSim

In this section, we measure the Pearson correlation coefficient (Pearson's $r$) and the Spearman's rank correlation coefficient by dataset (Spearman's $\rho$) between the LEMUR estimates and the exact MaxSim similarities, averaged over queries. Table 4 shows the correlation coefficients by dataset. The correlation coefficients are high ($\geq 0.942$) on each dataset.

*Table 4.* Pearson correlation coefficient and Spearman's rank correlation coefficient by dataset, averaged over queries.

| Dataset | Pearson's $r$ | Spearman's $\rho$ |
|---|---|---|
| HotpotQA | 0.966 | 0.959 |
| NQ | 0.952 | 0.942 |
| Quora | 0.979 | 0.972 |
| Touche-2020 | 0.963 | 0.951 |
| TREC-COVID | 0.994 | 0.990 |
| FiQA | 0.971 | 0.964 |
| SCIDOCS | 0.969 | 0.961 |

## E. Long Queries

While the default number of query vectors used in ColBERTv2 is 32, for long queries, e.g., in agentic applications, it can be preferable to use more vectors. To test the effectiveness of LEMUR with longer queries, in the figure below, we use the ArguAna dataset from BEIR with 300 vectors per query, following Santhanam et al. (2022b). LEMUR maintains its effectiveness in both retrieval performance and MaxSim approximation quality (Pearson correlation coefficient 0.987, Spearman's rank correlation coefficient 0.983).

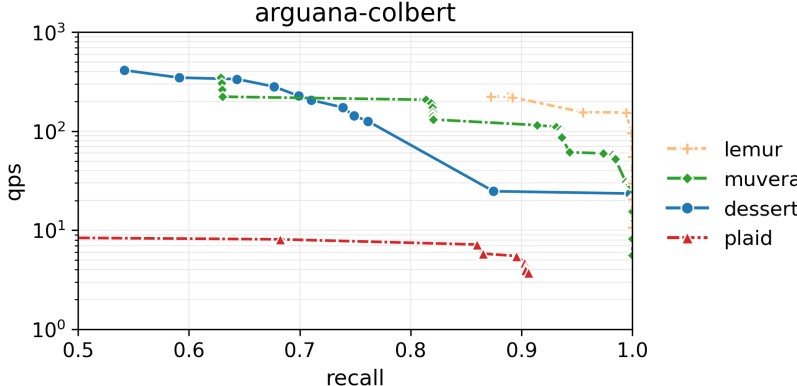

# F. Ablation Study Additional Results

## F.1. Additional Results for $k = 100$

In this section, we present additional results for the embedding dimension ablation study on eight BEIR datasets for $k = 100$. For discussion, see Section 6.2.

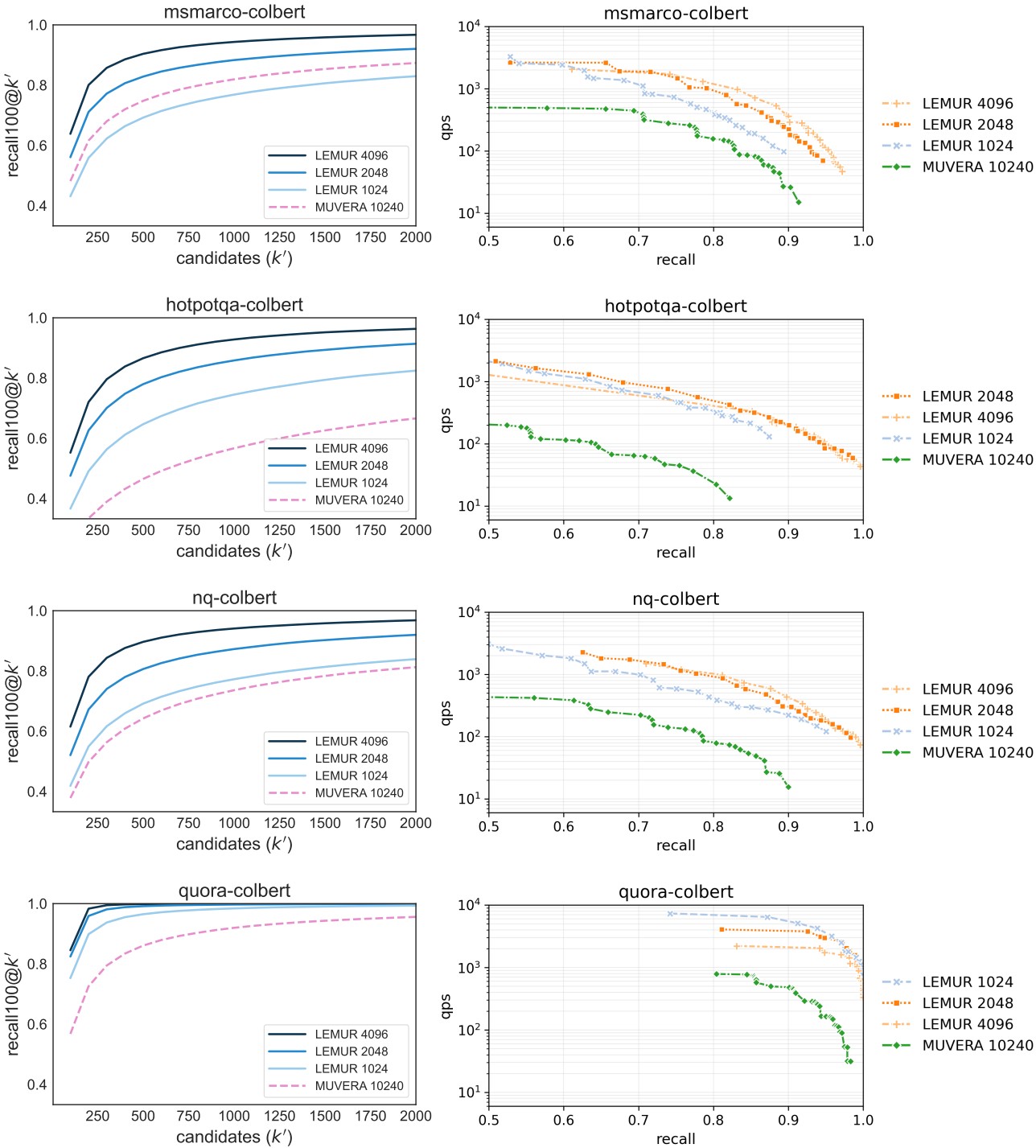

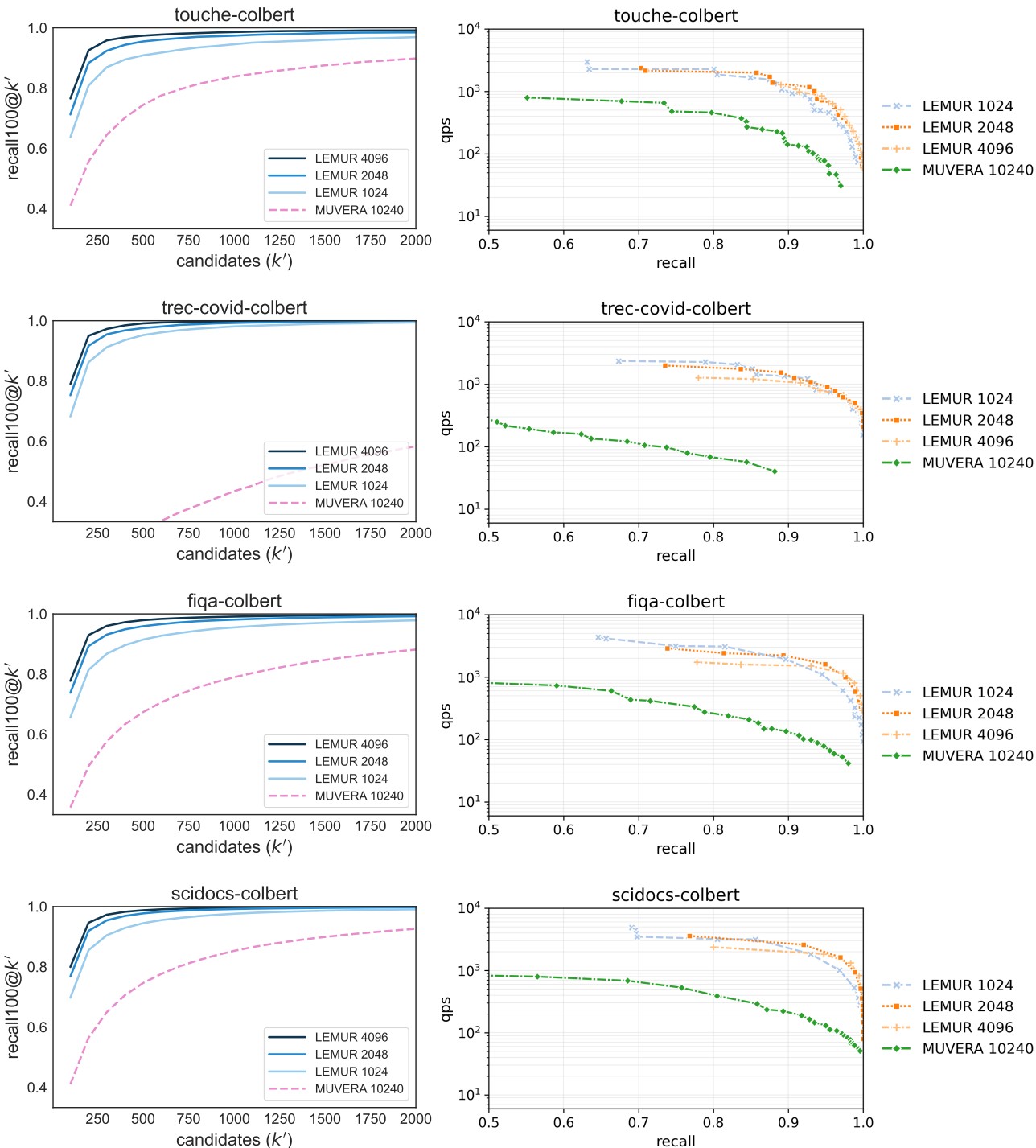

## F.2. Additional Results for $k = 10$

In this section, we present additional results for the embedding dimension ablation study on eight BEIR datasets for $k = 10$. The differences in recall between the configurations are similar to those for $k = 100$.

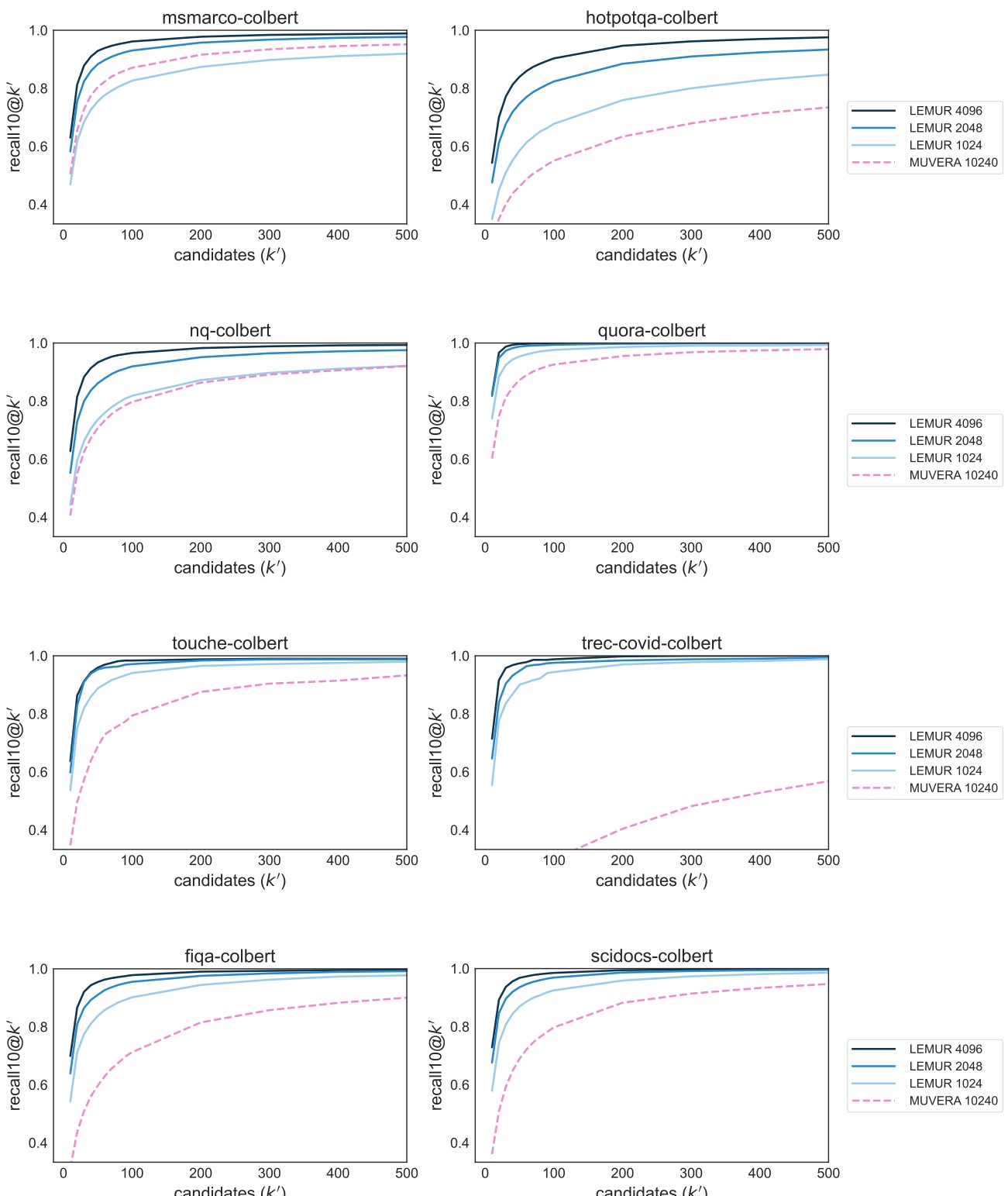

# G. End-to-End Performance Additional Results

## G.1. ColBERTv2

In this section, we present the complete end-to-end performance results on eight BEIR datasets. On all eight datasets, LEMUR significantly outperforms the baseline methods. For further discussion, see Section 6.3.

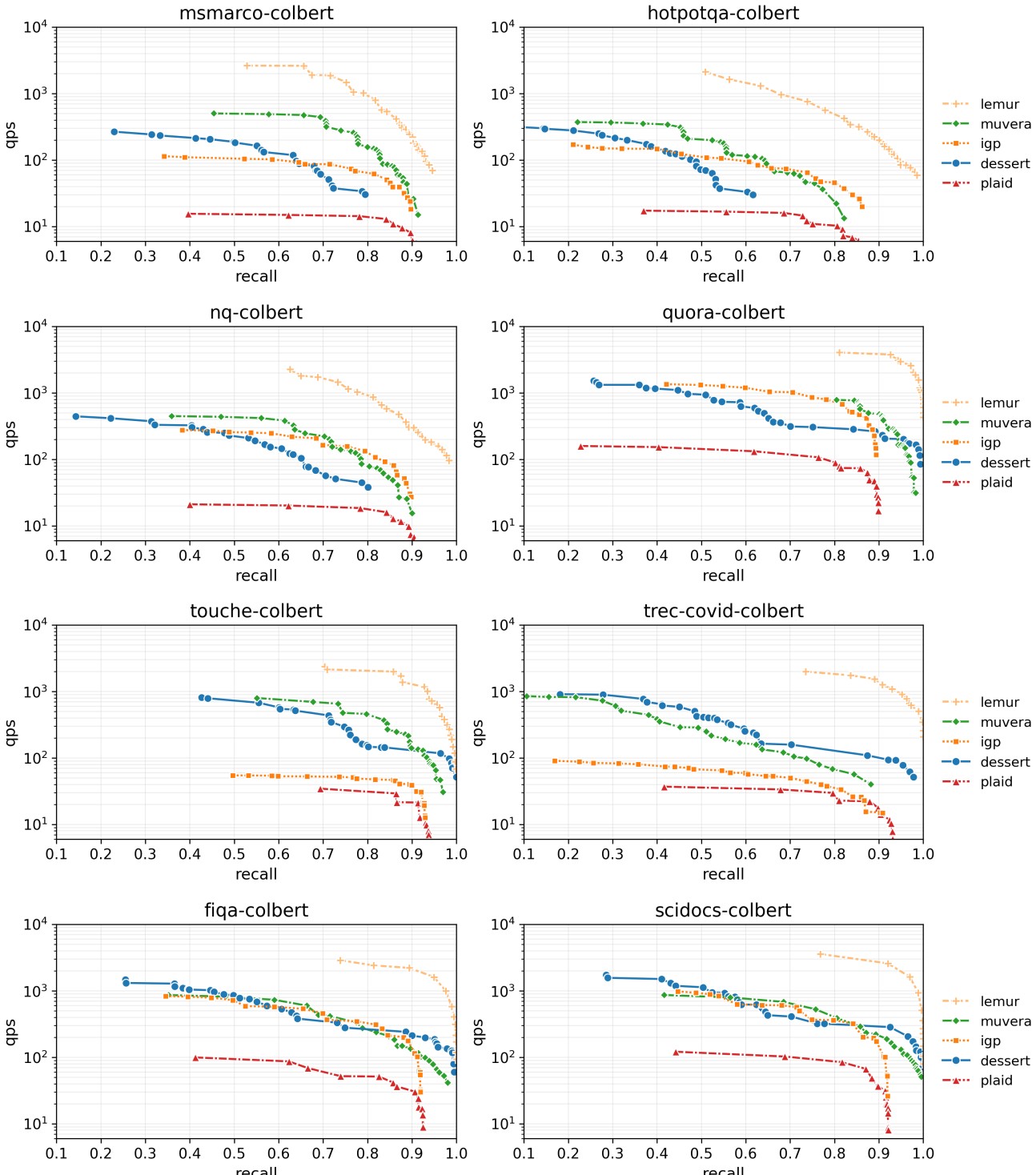

## G.2. Other Multi-Vector Embedding Models on SCIDOCS

In this section, we perform the experiment in Section 6.3 using the same four modern multi-vector embedding models on the SCIDOCS dataset. In addition, we also present results using the LateOn[2] and mxbai-edge-colbert-v0-32m[3] models. The results are similar to those for the Quora dataset.

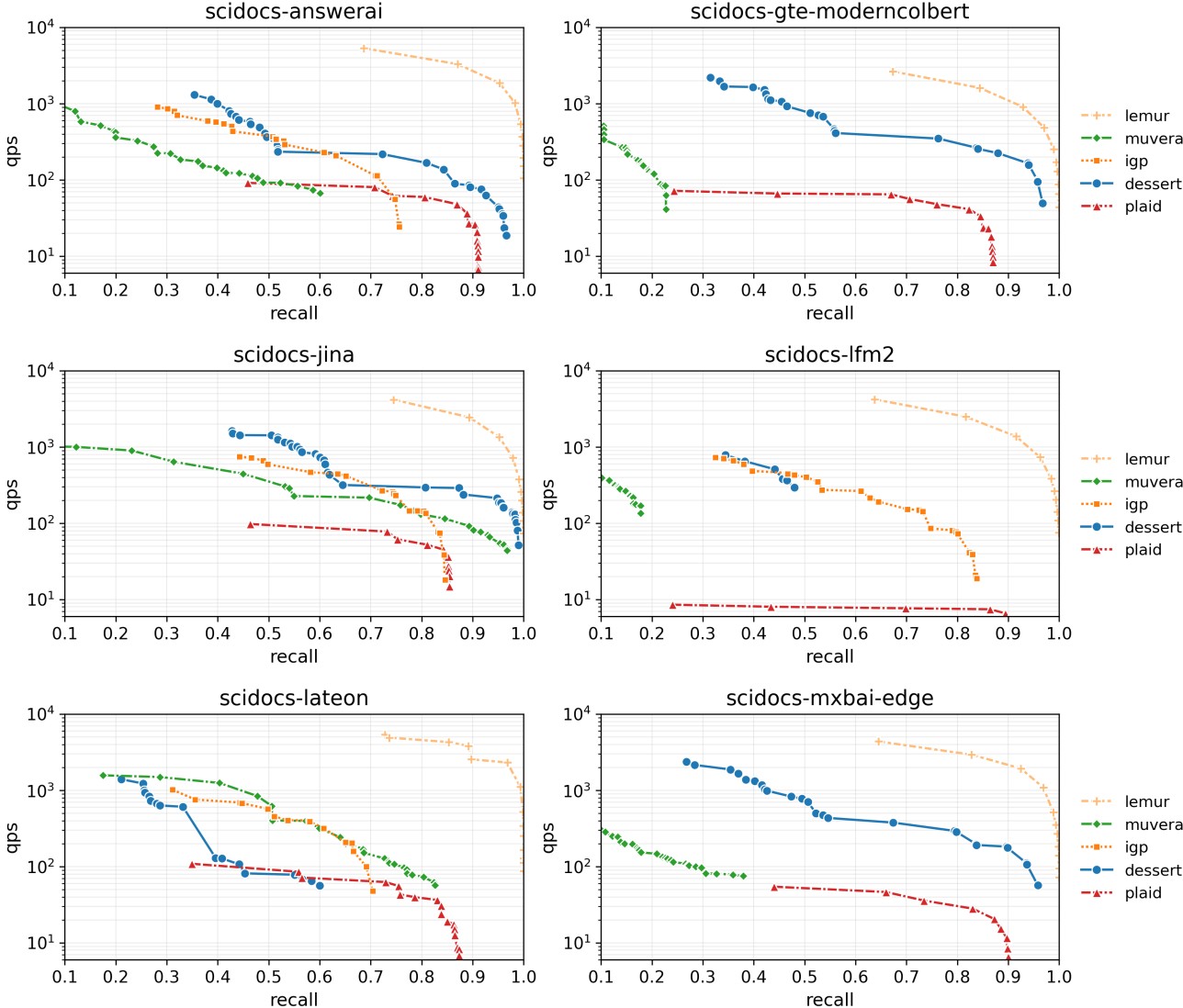

---

[2] https://huggingface.co/lightonai/LateOn
[3] https://huggingface.co/mixedbread-ai/mxbai-edge-colbert-v0-32m

# H. Effect of Query Distribution

## H.1. Document Encoder

In this section, we repeat the end-to-end performance experiments of Section 6.3 with LEMUR trained using a subset of the corpus $C_1, \ldots, C_m$ (encoded using the document encoder $D$) directly as a training set. This training method requires neither additional data nor access to an encoder. LEMUR still significantly outperforms the baseline methods.

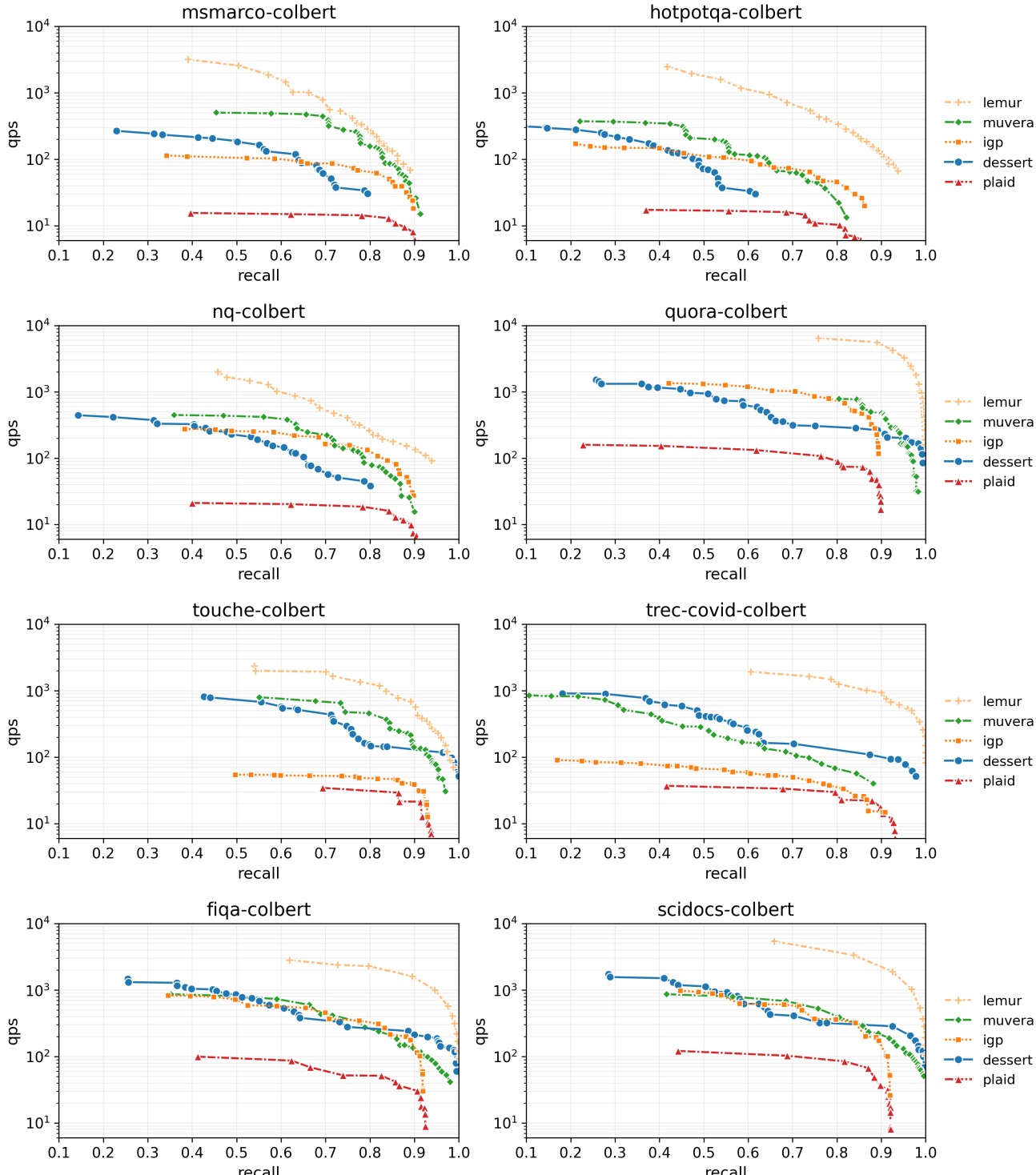

## H.2. Training Set Selection

In this section, we study the effect of using a separate sample of actual queries for training LEMUR. In the figures below, we consider LEMUR trained using three different training sets (see Section 4.3):

- `lemur (query)`: a separate sample of training queries, encoded using the query encoder $Q$

- `lemur (corpus-query)`: a sample of embeddings from the corpus documents encoded using the query encoder $Q$ (the default used in this paper, not available for visual document retrieval models)

- `lemur (corpus)`: a sample from the corpus token embeddings (encoded using the document encoder $D$)

On both HotpotQA and ViDoRe, using a sample of actual queries yields a consistent performance improvement for LEMUR. However, LEMUR is robust to the choice of training data.

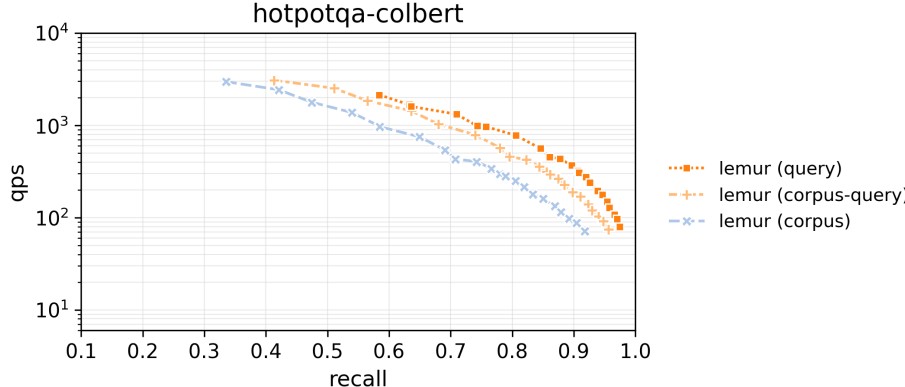

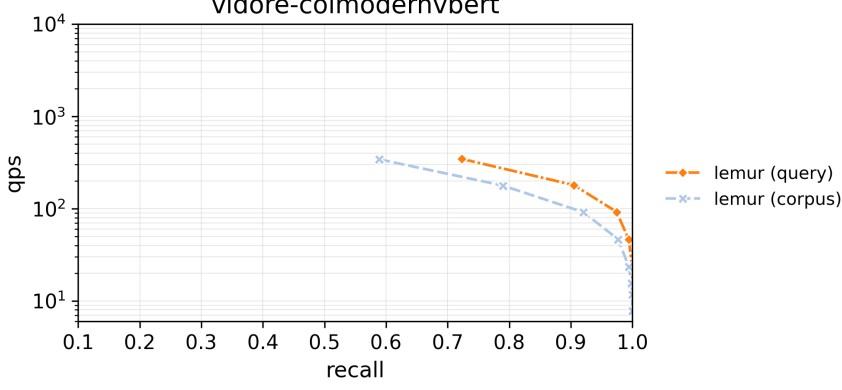

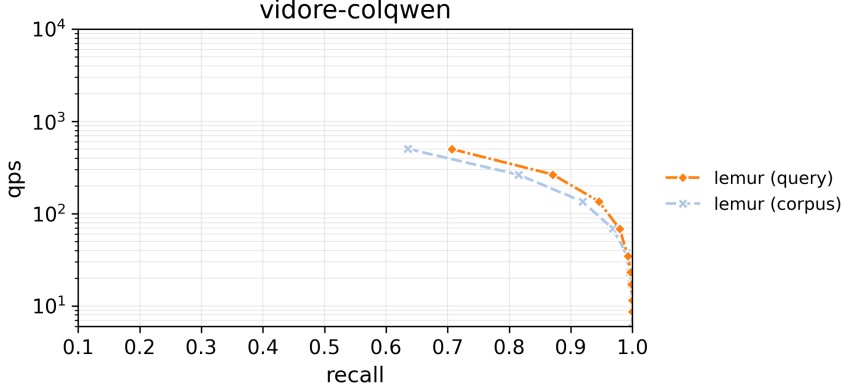

### H.3. Random Hidden Layer Weights

In this section, we perform an additional experiment in which we do not train the feature encoder $\psi$ at all. Instead, we use an MLP with *random weights* in the hidden layer, so the only component that adapts to the training data is the final projection layer. Specifically, we sample $R \in \mathbb{R}^{d' \times d}$ with independent standard normal entries and use random features

$$\psi_{\text{ELM}}(x) = \sqrt{\frac{2}{d'}}\, \text{GELU}(Rx).$$

This is often referred to as an extreme learning machine (ELM). The figure below shows the performance of the ELM version of LEMUR compared to the version with a trained MLP on the NQ dataset. For discussion, we refer to Section 4.3.

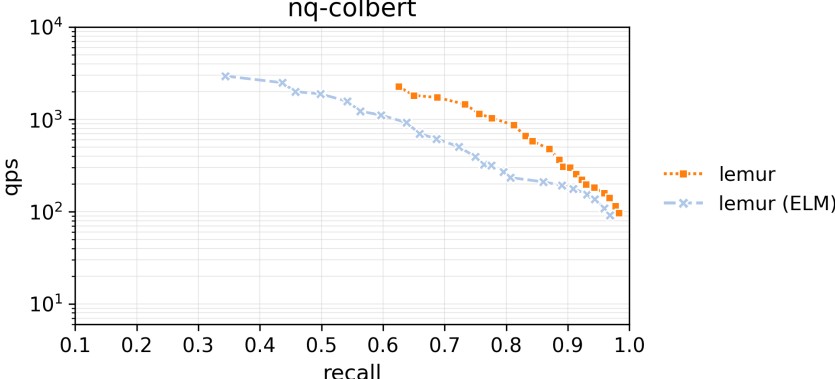

