# OpenReview forum: "LEMUR: Learned Multi-Vector Retrieval"
_ICML.cc/2026/Conference — ICML 2026 regular_

### Official Review · Reviewer_oHHJ · 2026-03-10

**Soundness:** 3
**Presentation:** 3
**Significance:** 3
**Originality:** 3
**Overall Recommendation:** 4
**Confidence:** 3

**Summary:**

Multi-vector retrieval boasts superior semantic matching performance, and it has thus attracted widespread attention in the information retrieval field. However, its latency issue has hindered its large-scale practical deployment. This paper proposes a new Learned Multi-Vector Retrieval framework called LEMUR to address the critical latency issue of multi-vector retrieval.

To this end, LEMUR reduces multi-vector similarity search to a single-vector ANNS (Approximate Nearest Neighbor Search) task through two consecutive steps:
First, casting the original similarity approximation as a supervised multi-output regression problem solvable by a lightweight one-hidden-layer MLP.
Second, transforming the model’s inference into maximum inner product search in a learned latent space, enabling seamless integration with well-optimized single-vector ANNS libraries.

Extensive experiments on six BEIR text datasets, 1 ViDoRe visual document dataset, and 7 multi-vector models (5 text, 2 visual) show that LEMUR outperforms SOTA multi-vector retrieval methods  by an order of magnitude in speed, achieves higher recall with far lower-dimensional embeddings (1024D LEMUR > 10240D MUVERA FDE), and demonstrates strong robustness across different training data and model types.

The framework is open-sourced and bridges the latency gap between single- and multi-vector retrieval, unlocking the scalability of multi-vector models for real-world applications.

**Compliance With Llm Reviewing Policy:**

Affirmed.

**Final Justification:**

The authors' rebuttal has fully resolved my questions about the robustness of the method in this work.

**Key Questions For Authors:**

1. LEMUR shows strong training robustness across diverse data sources, but the underlying mechanisms are not analyzed. Could you elaborate on the key factors driving this robustness, and is there a bound for its valid scope?
2. You use a fixed sum pooling strategy for query vectors, yet ignore variable query length impacts. Have you tested LEMUR on extreme-length queries?

**Limitations:**

Please give a detailed explanation of LEMUR's robusness among different datasets.

**Strengths And Weaknesses:**

Strengths:
1. The two-step reformulation converts the computationally expensive multi-vector MaxSim search into a supervised learning task and then single-vector ANNS, which can leverage mature single-vector retrieval infrastructure without sacrificing accuracy.
2. LEMUR outperforms all SOTA baselines by 5–11x in QPS  at ≥80% recall, achieves higher recall with much lower-dimensional embeddings, and generalizes excellently to both text and visual multi-vector retrieval tasks.
3. The training of LEMUR is robust. It maintains high performance with diverse training data sources (actual queries, corpus encoded by query/document encoders) and even works well without dedicated query training sets or additional encoders, lowering practical deployment barriers.
4. The paper is well written and can be clearly understood.

Weaknesses:
1. While the training process in LEMUR exhibits strong robustness, the underlying reasons for this robustness are not analyzed, making it rather difficult to comprehend its generalization mechanism across different datasets.
2. The paper does not investigate the impact of variable query lengths (e.g., short one-word queries vs. long multi-sentence queries) on LEMUR’s retrieval performance. It is unclear whether the fixed pooling strategy (sum of token latent embeddings) is optimal for all query lengths, and no targeted optimization is proposed.

---

> ### Author Rebuttal · Authors · 2026-03-30
>
> We thank the Reviewer for their helpful feedback and questions.
>
> **Robustness of LEMUR**: LEMUR is indeed robust w.r.t. the choice of its training data. The results of Appendix D show that an accurate model can be trained by using either 1) queries encoded by query encoder; 2) documents encoded by a query encoder; or even 3) documents encoded by a document encoder. We hypothesize that this is because both query and document tokens are by design mapped by the late interaction model into a common semantic embedding space. As a result, while the used training sets are different, they all provide samples from roughly the same geometry.
>
> We performed an additional experiment where we did not train the feature encoder $\psi$ at all, i.e., we used an MLP with random weights in the hidden layer (often called an extreme learning machine, ELM) and thus the only part that adapts to the training data is the final projection layer. While the performance clearly degrades compared to training the feature encoder, even this version outperforms the baseline methods. If LEMUR still works well in this scenario, it suggests its performance does not primarily depend on learning a highly training data-specific representation of queries. Instead, the hidden layer mainly serves to expand the input into a richer feature space. Thus the different training data may slightly change the distribution of examples, but do not change the underlying geometry the linear layer is fitting to.
>
> **Effect of query length:** The argument retrieval data set ArguAna used in our experiments contains longer multi-sentence queries: its queries have on average 193 words, whereas on the other datasets used in our experiments the average query length is between 6 and 18 words. We realized that in our original experiments we have used the ColBERTv2 default of 32 embeddings per query also for ArguAna, and represent below the results (best QPS at >= 80\% recall) using 300 embeddings per query in line with the original ColBERTv2 paper. LEMUR outperforms the baselines also on the longer multi-sentence queries.
>
> | Method | LEMUR | MUVERA | DESSERT | PLAID |
> |--------|-------|--------|---------|-------|
> | QPS    | 156   | 115    | 23      | 5     |

---

> > ### Author Rebuttal · Reviewer_oHHJ · 2026-04-03
> >
> > I appreciate the authors’ response. Having reviewed the revisions and clarifications, I consider the robustness of the proposed method acceptable, and I hereby raise my originality score from 2 to 3.

---

> > > ### Author Response · Authors · 2026-04-04
> > >
> > > We thank the Reviewer for their helpful feedback, which strengthened our paper. Since we believe that we have fully addressed the concerns raised, we hope that the Reviewer might also consider updating their overall score.

---

### Official Review · Reviewer_BB2C · 2026-03-11

**Soundness:** 2
**Presentation:** 3
**Significance:** 3
**Originality:** 2
**Overall Recommendation:** 5
**Confidence:** 4

**Summary:**

This paper proposes `LEMUR`, a framework for efficient multi-vector retrieval. The key idea is to for every document in a dataset, map its multi-vector representation into a single-vector representation and jointly learn a function which maps a query's multi-vector representation into a single-vecto representation. This allows the method to use existing single-vector ANNS libraries for efficient retrieval. The paper evaluates the method on several multi-vector text models, and visual document retrieval models, comparing against MUVERA, DESSERT, IGP, and PLAID.

**Compliance With Llm Reviewing Policy:**

Affirmed.

**Final Justification:**

The authors have resolved all my concerns in the rebuttal. I am raising the overall score from 4 to 5.

**Key Questions For Authors:**

1. Can the authors quantify more explicitly the cost of training and updating the corpus-specific learned reduction, and explain how this affects deployment in dynamic corpora?
2. Can the authors give more intuition or analysis for why a shallow MLP plus an inner-product latent space is expressive enough to approximate `MaxSim` well in practice?
3. Can the authors report more direct approximation-quality measurements between the learned single-vector scores and the exact `MaxSim` scores, for example relative error or correlation statistics?
4. Can the authors position `LEMUR` more explicitly relative to `WARP`, ideally with a direct discussion or comparison where feasible?

**Limitations:**

Yes.

**Strengths And Weaknesses:**

Strengths:

- The paper proposes a neat reduction from multi-vector retrieval to single-vector ANNS.
- The empirical evaluation is broad. The method is tested on several modern text and visual multi-vector models.
- The end-to-end performance gains are strong.
- The paper includes helpful ablations on latent-space dimension. The implementation details are also clear.

Weaknesses:

- The method requires a corpus-specific learned reduction. While the paper discusses scalable training, the cost of training and updating the learned reduction is not sufficiently discussed and compared.
- The paper hardly explains why the method could work. I was curious why the paper can approximate functions as complex as MaxSim simply with a two-layered MLP along with inner product.
- It is suggested to perform more measurements to show the difference between the scores produced by the single-vector representation and the ground-truth scores produced by MaxSim. I would suggest to at least measure relative error to better support the claims in the paper.
- It would be better if the authors can discuss and compare with WARP (https://arxiv.org/pdf/2501.17788)  in multi-vector search.

---

> ### Author Rebuttal · Authors · 2026-03-30
>
> We thank the Reviewer for constructive and detailed feedback, especially suggestions for explaining the intuition behind the proposed method and for discussing the index building and index updating times in more detail helped to improve the manuscript.
>
> **Cost of training the corpus-specific learned reduction:** On our largest dataset, MS MARCO, training the feature encoder $\psi$ took 16 minutes on CPU, and computing the weight matrix $W$ of the final projection layer by linear regression took 75 minutes (2000 documents per second). Training time of the feature encoder $\psi$ is approximately constant regardless of the corpus size, whereas the computing time of $W$ scales linearly with the number of documents.
>
> New documents can be indexed quickly by computing the document weights by linear regression and adding them to the weight matrix $W$, since the feature encoder is kept fixed. This means that updating the LEMUR index does not add significant computational overhead, since in practice the cost of adding documents to the retrieval system is dominated by the cost of computing the document embeddings. While our results indicate that LEMUR is robust w.r.t. the choice of training data (see Appendix D), it can be advisable to retrain the whole index periodically if a lot of new data is added to ensure optimal performance in view of a possible distribution shift. This requirement of periodical re-indexing applies also to the baseline clustering-based methods and also commonly to single-vector retrieval methods. Compared to the baseline methods, LEMUR has the fastest indexing for large datasets (see our answer to Reviewer 5AgD for details).
>
> **Intuition of why a two-layer MLP is sufficient for estimating MaxSim**: Our intuition is that the MLP does not estimate the MaxSim function in one shot, as discussed in Section 3.1 of the manuscript. The MaxSim similarity between the query and the $l$th document is a sum
> $$\mathrm{MaxSim}(X,C_l) = \sum_{x \in X} \underset{c \in C_l}{\max} \langle x,c\rangle,$$
> and instead of the whole sum, the MLP in LEMUR estimates its components
> $$g_l(x) := \underset{c \in C_l}{\max} \langle x,c\rangle,$$
> which is a much simpler task. Crucially, these components are only estimated for a _fixed_ set of corpus documents $C_l$ and each $g_l$ is a convex piecewise-linear function of $x$ which is precisely a good fit for a wide, shallow MLP. We will clarify this explanation in the final version of the manuscript.
>
> **MaxSim approximation quality:** We performed an additional experiment where we measure the Spearman rank correlation between the LEMUR estimates (at the latent space dimension $d' = 2048$ used in all of our experiments) and the true MaxSim scores, averaged over the queries. The estimates generated by LEMUR indeed correlate highly with the MaxSim scores: the correlations were over 0.94 on all datasets (see table below). We will include these results in the final version of the manuscript.
>
> | Dataset | ArguAna | SCIDOCS | Quora | NQ | HotPotQA | MSMARCO |
> |---|---|---|---|---|---|---|
> | Correlation | 0.981 | 0.960 | 0.971 | 0.940 | 0.959 | 0.951 |
>
> **Comparison to WARP:** WARP is a multi-vector retrieval engine designed specifically for multi-vector models trained using the XTR objective [1]. In our experiments, we consider only widely-used late interaction models, none of which use this objective. However, WARP contains improvements to the PLAID retrieval engine that it is based on, which may make it more efficient for models trained without the XTR objective as well. Therefore, we performed an additional experiment using WARP as a baseline. However, we were unable to achieve a high recall on any of the datasets despite considering a wide hyperparameter grid. On the SCIDOCS and Quora datasets at >= 50\% recall, WARP achieves best QPS of 987 and 586, respectively, which makes it faster than PLAID but still slower than the other baseline methods.
>
> [1] Lee et al. Rethinking the role of token retrieval in multi-vector retrieval. NeurIPS (2023)

---

> > ### Author Rebuttal · Reviewer_BB2C · 2026-04-03
> >
> > Thanks for your reply. I am still concerned about some of the issues.
> > 1. **Cost of training the corpus-specific learned reduction**: the cost of training the model looks pretty high. It is suggested to explicitly discuss the limitation in the paper and compare it with the counterpart in the baseline methods. If I understand correctly, with such heavy retraining costs, the method can hardly support workloads with updates.
> > 2. **Intuition of why a two-layer MLP is sufficient for estimating MaxSim**: I understand that this is an open question. However, I am unconvinced of the explanations. Could the authors please provide more explanations, especially on why MLP can fit MaxSim well when the # of parameters in MLP is much smaller than the # of parameters in the raw datasets?
> > 3. **MaxSim approximation quality**: The authors did not reply to my question directly. As the authors claim that their algorithm can approximate the scores of MaxSim well. Please report the error of the approximate scores (compared with the true scores). Spearman rank correlation cannot support the paper's claim.

---

> > > ### Author Response · Authors · 2026-04-04
> > >
> > > We thank the Reviewer for their feedback and continued interest.
> > >
> > > 1. As stated in our rebuttal, compared to the baseline methods, LEMUR in fact has the fastest indexing times and is therefore the most scalable method (please see our answer to Reviewer 5AgD for more detail). LEMUR is also well suited for index updates as explained in our rebuttal: for example, on MSMARCO, new documents can be indexed at a rate of 2000 documents per second. Reindexing is a standard procedure that is also required by the baseline methods, and should only be applied occasionally in case of significant data drift; this is typically done in the background. The cost for reindexing is much lower for LEMUR than for the baseline methods.
> > >
> > > 2. In our rebuttal, we have explained that the MLP in LEMUR does not estimate the full MaxSim score at once, but instead learns simpler fixed-document components that are convex, piecewise-linear, and therefore well suited to a wide, shallow MLP. We believe this is the intuitive explanation why a shallow MLP works. However, there might still be a slight misunderstanding regarding the number of parameters in the MLP. The number of parameters is in fact large as the final linear transformation $W$ has one row of length $d'$ for each document in the corpus. This means that, for example, for MSMARCO, the number of parameters in the MLP is roughly 24% of the size of *all* the corpus token embeddings. The key to the efficiency in our method comes from the reduction to approximate nearest neighbor search (please see Section 3.2 of the paper) which means that the large final linear transformation does not need to be computed explicitly.
> > >
> > > 3. In our rebuttal, we reported Spearman rank correlations because they explain best why our method works so well in practice as only the ordering matters for generating the candidate set. We appreciate that the Reviewer would still like to see the relative errors, so we have provided the mean relative error for each dataset in the table below. The error is small for each dataset.
> > >
> > > | Dataset        | ArguAna | SCIDOCS | Quora | NQ    | HotPotQA | MSMARCO |
> > > |----------------|---------|---------|-------|-------|----------|---------|
> > > | Relative error | 0.024   | 0.034   | 0.031 | 0.045 | 0.037    | 0.040   |
> > >
> > > For completeness, in the table below, we also report the Pearson correlation coefficients per dataset:
> > >
> > > | Dataset     | ArguAna | SCIDOCS | Quora | NQ    | HotPotQA | MSMARCO |
> > > |-------------|---------|---------|-------|-------|----------|---------|
> > > | Correlation | 0.985   | 0.968   | 0.979 | 0.952 | 0.966    | 0.958   |

---

### Official Review · Reviewer_5AgD · 2026-03-13

**Soundness:** 4
**Presentation:** 3
**Significance:** 3
**Originality:** 3
**Overall Recommendation:** 4
**Confidence:** 5

**Summary:**

The paper proposes Learned Multi-Vector Retrieval (LEMUR) a method to approximate MaxSim similarity by first framing the multi-vector retrieval problem as a multi-output regression problem where given a set of query documents $X$ the regression model directly outputs scores for each of the $m$ documents in the corpus. To make the regression scalable, it is formulated so the final score calculation is represented as an inner product between a single vector for the query and vector representations for each document in the corpus. This formulation allows existing Approximate Nearest Neighbor (ANN) algorithms to be used to retrieve a candidate set of documents efficiently which are then reranked by the full MaxSim similarity. The results of a detailed set of evaluations show LEMUR beats many other methods in effectiveness and efficiency.

Overall, I feel the paper provides a notable improvement on prior common approaches most notably PLAID and MUVERA, and provides a unique approach to improve retrieval speed for late-interaction retrieval models. My main gripe with the paper is that a simple and conceptually very similar baseline is missing; reranking retrievals from a dense retrieval model with ColBERT. Clearly LEMUR has improvements over this method, most notably that it does not require re-encoding the whole corpus twice, but it seems like an important baseline to consider given the simplicity compared to LEMUR and some simple approaches to remedy the limitations. If this baseline is added and is clearly beaten by LEMUR I would be open to a stronger recommendation.

**Compliance With Llm Reviewing Policy:**

Affirmed.

**Key Questions For Authors:**

1. How does LEMUR compare to retrieving with a strong dense retrieval model and reranking with ColBERT?
2. What are the benefits of LEMUR if you could learn a dense retrieval model that would produce the same representations? Is there a reason to think a dense retrieval model couldn’t produce these representations?

**Limitations:**

Yes

**Strengths And Weaknesses:**

Strengths:
+ The formulation of turning MaxSim (or maybe any retrieval task) into a multi-output regression problem is unique (as far as I am aware) and could be useful in other retrieval tasks as well.
+ LEMUR does not require any query data to train and can just use the documents as pseudo-queries to train the model.
+ The results of LEMUR are far better than other major approaches to speed up MaxSim similarity such as PLAID or MUVERA.
+ The method is tested on a variety of datasets which helps show that the method can generalize.
+ The ablations are sensible and cover important aspects of the proposed method.
+ The authors use multiple late-interaction models that include models from different modalities which shows the generalizable nature of LEMUR.


Weakness:
- Each corpus needs to have a specific model trained for it, though this could be seen as similar to an “indexing” phase in other late-interaction frameworks such as PLAID.
- No theoretical proof that LEMUR is optimal or can match MaxSim.
- The work does not seem to address a key question, why can’t a dense retrieval system learn the LEMUR representations directly? If a dense retrieval system could, what advantage would LEMUR have then?
- One baseline that I think is directly comparable to LEMUR which is not tested is simply using an off-the-shelf single-vector embedding model to act as a retriever and rerank with ColBERT. The efficiency would be identical to LEMUR, but would not require learning a separate model although it would require encoding the whole collection twice. Still, I believe this baseline would be useful and help better understand the trade-offs of LEMUR.
- If you did not want to do encoding twice, a very simple approach could also be just average the token embeddings to produce a single vector representation. There have also been prior models that jointly train both capabilities which could be an alternative.
- Similarly, using BM25 would be interesting.
- In “A Reproducibility Study of PLAID” the authors find that using an adaptive reranking approach with BM25 has strong performance. Perhaps it is worth considering as a baseline.
- Results on more complex datasets such as BRIGHT (BRIGHT: A Realistic and Challenging Benchmark for Reasoning-Intensive Retrieval) or CRUMB (Benchmarking Information Retrieval Models on Complex Retrieval Tasks) would be useful to understand if LEMUR is still capable beyond fairly straightforward retrieval scenarios.
- There does not seem to be any information about the index or indexing. Specifically, how long does indexing take with LEMUR versus other approaches, specifically PLAID and MUVERA? And how much space does it take compared to these? PLAID has the benefit of also compressing the embeddings while LEMUR does not directly show this given the
- This line “increasing $d’$ from 1024 to 2048 significantly improves latency” was a bit confusing for me, I would expect larger embedding dimensions to decrease latency, but the opposite is true. I assume this is at a specific recall value. In any case, it would be useful for the authors to expand on why this is the case.
- No standard IR metrics on the datasets tested. I understand this isn’t always common for work in efficient retrieval so I don’t think it is a big negative, but it would help to understand how the various approaches impact the final outcome.
- It would be interesting to know if there was a way to learn a single-vector document representation $w_i$ directly from the ColBERT representation $C_i$ instead of requiring corpus specific training.
- There doesn’t seem to be a proper results section, it is blended with the experiments section and the end-to-end results don’t seem to be discussed.
- Related, the structure of the experiments section feels a bit weird, as the ablation study comes before the “End-to-end performance” section and the “End-to-end performance” section seems to not include any information about the results of the performance, it just includes information about the models used. Additionally, the ablation section reads more like a result section while the rest of the “Experiments” section gives information about the experiments such as models used and experimental settings.
- I would suggest moving some of the experiment details to the appendix and shortening the discussion section to have a more standard results section.


Specific Comments:
1. On line 175, $g_l$ is introduced without connecting it to $g$. It would be useful to understand how they connect.
2. The paper “Scaling Laws for Embedding Dimension in Information Retrieval” may be relevant to justify the results shown in the ablation with varied embedding dimensions.

---

> ### Author Rebuttal · Authors · 2026-03-30
>
> We thank the Reviewer for the helpful and detailed feedback. For clarity, we only address the concerns we consider most relevant. If there are still any outstanding issues that require clarification, we are happy to provide more details.
>
> **Comparison to retrieve-and-rerank using a strong single-vector retriever:** Note that the task we consider in this article is efficient multi-vector search in task-agnostic fashion, also considered by the baseline methods like MUVERA, and without assuming that there are other pre-trained models available. Thus, we do not consider multi-stage IR pipelines. If one wants to use a multi-vector retriever trained on a specific domain, a corresponding single-vector retriever would additionally need to be trained and this training is typically more difficult. For multimodal retrieval, such as visual document retrieval models considered in the paper, this is even more difficult and less likely to work. This highlights the need for special multi-vector search indexes, and LEMUR is superior to prior methods, including a previous single-vector approximation method MUVERA that has been integrated into state-of-the-art vector databases.
>
> However, we acknowledge that the requested comparison may be interesting for practical purposes. In contrast to using a strong single-vector retriever, LEMUR is directly aligned with the MaxSim objective and does not need to pay the double encoding cost. We have performed an additional experiment where we compare LEMUR to retrieving candidates with a single-vector retrieval model and re-ranking using MaxSim. We consider a baseline suggested by the Reviewer and two strong single-vector embedding models: (1) single-vector representations computed by averaging the token embeddings; (2) nomic-embed-text-v1.5; (3) jina-embeddings-v5-text-nano. To reflect Table 2 in the article, the average QPS for SCIDOCS and HotpotQA at Recall >= 80% can be found below. These measurements *do not include the time it takes to compute the single-vector embedding of a query*, indicating that LEMUR needs to rerank much fewer candidates since it directly approximates MaxSim.
>
> | | LEMUR | RR (AVG) | RR (Nomic) | RR (Jina) |
> |---|---|---|---|---|
> | SCIDOCS | 3353 | 410 | 1065 | 583 |
> | HotpotQA | 425 | -- | 46 | 257 |
>
> For the benefits of LEMUR in comparison to trying to directly learn single-vector representations, please see our answer to Reviewer J12w.
>
> **Information about the index and indexing:** In addition to having the fastest query times, LEMUR also has faster indexing compared to the baseline methods. On our largest dataset, MS MARCO, the total indexing time on CPU was 4.7 hours which is composed of MLP training (16 minutes), computing the LEMUR features of all documents (75 minutes), and building the ANN graph (192 minutes). Computing the LEMUR features scales linearly with the number of documents, while the ANN index construction is typically $O(n \log n)$ for graph indexes. In comparison, the ANN graph construction alone took over 12 hours for MUVERA due to the much higher dimensionality $d ' = 10240$ of its single-vector representations. Building the index using only a CPU is feasible only for LEMUR and MUVERA. The other baseline methods, including PLAID, scale poorly to large corpora since they require clustering the collection of all the document embeddings. For instance, on MS MARCO this means clustering 600 million document embeddings into 264k clusters; this would take days on a CPU, and still takes over an hour using a GPU.
>
> LEMUR has lower storage cost than MUVERA since a lower-dimensional single-vector representation is sufficient to obtain accurate results when LEMUR is used. The storage cost of LEMUR is higher than that of PLAID since we have not considered compression of the embeddings. We think this is interesting future work and have mentioned this limitation in the manuscript (end of Section 7).
>
> **Information retrieval metrics:** We have performed an additional experiment measuring the standard information retrieval metric MRR@10 on MSMARCO and NDCG@10 on other BEIR datasets. For the results, see our answer to Reviewer J12w.
>
> **Presentation:**
>
> - As the Reviewer suspects, “increasing $d'$ from 1024 to 2048 significantly improves latency” indeed means improving latency at the fixed recall level. This is because even though a larger latent space dimension increases the computational cost of ANN queries, it also leads to more accurate MaxSim estimates that enable selecting a smaller candidate set that is re-ranked.
>
> - Our definition of $g':\mathbb{R}^d \rightarrow \mathbb{R}^{m'}$ as $g = g_{I'}$ is indeed ambiguous notation. The function $g'$ picks the components indexed by the set $I' \subset [m]$ from the components of $g(x) = (g_1(x), \dots, g_m(x))$. We will update the formula to a more explicit form.
>
> - Thank you for the suggestions about the structure of the experimental section, we will improve it in the final revised version of the article.

---

> > ### Author Rebuttal · Reviewer_5AgD · 2026-04-04
> >
> > Thanks for your response. I think your response helps. I still think this paper has values and can be accepted. I believe my weak accept rating is still relevant as some questions about theoretical guarantees are left unresolved.

---

> > > ### Author Response · Authors · 2026-04-04
> > >
> > > We thank the Reviewer for their support and feedback, which helped improve the paper. We are glad that our response helped resolve the practical concerns. We appreciate that the Reviewer still has concerns about the absence of formal theoretical guarantees, which would certainly be valuable, but we trust that the community will still appreciate that the proposed method is both principled and empirically strong.

---

### Official Review · Reviewer_J12w · 2026-03-13

**Soundness:** 4
**Presentation:** 3
**Significance:** 3
**Originality:** 4
**Overall Recommendation:** 4
**Confidence:** 4

**Summary:**

This paper proposes LEMUR, a method that approximates the late-interaction scoring function used in ColBERT with a learned single-vector representation. Instead of performing MaxSim over query and document token embeddings at retrieval time, LEMUR learns a mapping that aggregates query token representations into a single vector and represents each document with a learned vector. Retrieval is then performed using efficient inner product search to obtain candidate documents, followed by optional MaxSim reranking.

The idea of approximating a multi-vector late-interaction retriever with a learned single-vector representation is interesting, and the paper presents a clear formulation of how the MaxSim scoring function can be approximated through a learned transformation. The paper is also well written and easy to follow. The motivation, derivations, and system design are generally clearly explained.

However, the experimental evaluation is not fully convincing, and several aspects of the approach would benefit from additional clarification. In particular, the paper mainly evaluates how well LEMUR approximates ColBERT’s MaxSim retrieval, rather than measuring standard IR effectiveness. In addition, it is not entirely clear how fundamentally different the proposed formulation is from learning a dense retriever that distills ColBERT scores.

**Compliance With Llm Reviewing Policy:**

Affirmed.

**Final Justification:**

The paper is interesting overall, hence I increased my score.

**Key Questions For Authors:**

1. What is the performance of MRR@10 in MSMARCO, NDCG@10 in TREC19 & 20

2. What is the zero-shot performance in BEIR?

**Limitations:**

yes

**Strengths And Weaknesses:**

**Strengths**

1. The idea is interesting. Approximating ColBERT-style late interaction retrieval using a single-vector representation is an interesting perspective and could potentially provide a useful efficiency–effectiveness trade-off.

2. The paper is well written and easy to understand. The formulation and motivation of the method are clearly explained.

**Weakness**
1. While the proposed formulation is interesting, it is not entirely clear how fundamentally different LEMUR is from learning a dense retriever that distills ColBERT scores.

In LEMUR, each query token embedding $x_i$ is mapped to a hidden representation using a learned function $\psi(\cdot)$, and the final query representation is obtained by aggregating token representations:

$$
q = \sum_{i} \psi(x_i)
$$

Document representations are learned as vectors $w_j$, and retrieval is performed using an inner product:

$$
\text{score}(q, d_j) = q^\top w_j
$$

This formulation appears very similar to a standard dense retrieval model, where a query encoder produces a single query embedding and retrieval is performed via inner product with document embeddings. In particular, one could imagine learning a dense retriever that directly distills the ColBERT MaxSim scores into such a representation.

Therefore, it would be helpful for the authors to clarify what the essential differences are between LEMUR and such a dense retrieval formulation, and why the proposed approach provides advantages beyond standard dense retrieval with knowledge distillation from ColBERT.

2. The experimental evaluation is not comprehensive.
The paper mainly evaluates how well LEMUR approximates ColBERT’s MaxSim retrieval, but it does not report standard IR metrics such as MRR@10 on MS MARCO or NDCG@10 on TREC DL 19/20, which are commonly reported in prior ColBERT work (e.g., ColBERTer).
It would also be useful to compare the zero-shot performance on BEIR benchmarks against results reported in prior work (e.g., Table 4 of https://arxiv.org/pdf/2302.07452
).
Without these evaluations, it is difficult to understand whether the proposed method preserves the actual retrieval effectiveness of ColBERT.

---

> ### Author Rebuttal · Authors · 2026-03-30
>
> We thank the reviewer for their helpful feedback.
>
> **Difference from learning a dense retriever that distills ColBERT scores:** We assume that the Reviewer refers to learning a dense embedding model, i.e., an encoder $D_{\mathrm{single}} : \Omega_{\mathrm{text}} \rightarrow \mathbb{R}^{d^\prime}$, that generates single-vector embeddings $\hat{x} := D_{\mathrm{single}}(\mathcal{X}) \in \mathbb{R}^{d'}$ and  $\hat{c} := D_{\mathrm{single}}(\mathcal{C}) \in \mathbb{R}^{d'}$ taking as input raw (e.g., in text form) queries $\mathcal{X} \in \Omega_{\mathrm{text}}$ and raw documents $\mathcal{C} \in \Omega_{\mathrm{text}}$. This encoder is learned using the MaxSim scores as training outputs, i.e, the objective is to learn a $d'$-dimensional presentation for which the inner products $\langle\hat{x},\hat{c}\rangle$ approximate the MaxSim similarities between the multi-vector representations of the queries and the documents.
>
> In contrast, LEMUR provides a more straightforward and computationally efficient approach by instead learning a _corpus-specific search reduction_ by formulating the MaxSim estimation as multi-output regression problem where the targets are the MaxSim similarities between the query and the $m$ documents. The LEMUR model does not take the documents as inputs; instead, the single-vector representations of the corpus documents are learned by linear regression as the rows of the weight matrix $W$ of the final projection layer, which is an easier task than learning a shared encoder $D_{\mathrm{single}}$ for both queries and documents. Instead of encoding raw queries $\mathcal{X}$ like a dense embedding model, LEMUR learns a single-vector representation of the queries by the encoder $\Psi: 2^{\mathbb{R}^d} \rightarrow \mathbb{R}^{d'}$, $\Psi(X) = \sum_{x \in X} \psi(x)$, that takes as an input an already encoded (by the multi-vector query encoder $Q$) query $X$.
>
> Since LEMUR operates on the queries and the documents that are already encoded by a multi-vector model, it does not require embedding the query and the document twice. Thus, it has fast training and inference: the query encoder of LEMUR is small (262k parameters), and its training time is 16 minutes on a CPU even on our largest data set (MS MARCO). In contrast, a dense retriever operates on text with much bigger encoders, and is much more difficult to train, since it must learn a generic relevance function rather than a corpus-specific reduction. This means that LEMUR can be used easily with any existing late interaction model, including multimodal visual document retrieval models, without expensive dense retriever training. Note that the problem we consider in this article is efficient multi-vector search in task-agnostic fashion and we do not assume that there are other pre-trained models available. LEMUR is superior to prior multi-vector search methods, including a previous single-vector approximation method MUVERA that has been integrated into state-of-the-art vector databases.
>
> **Evaluation using information retrieval metrics:** To verify that LEMUR maintains the effectivity of MaxSim retrieval, we present additional results in the table below. We measure the QPS of LEMUR and the baseline single-vector approximation method MUVERA at different MRR@10 for MS MARCO using ColBERTv2 embeddings. The MRR@10 of exact MaxSim search using ColBERTv2 embeddings is 39.9. In contrast, exact MaxSim search using ColBERTv2 embeddings has QPS $\approx$ 0.025.
>
> | MRR@10 | 39.9 | 39.8 | 39.4 | 39.2 | 39.0 | 38.8 | 38.6 |
> |--------|------|------|------|------|------|------|------|
> | LEMUR  | 44   | 48   | 125  | 230  | 259  | 453  | 587  |
> | MUVERA | --   | --   | --   | --   | 16   | 16   | 29   |
>
> Unfortunately, we were unable to perform new experiments on TREC during the rebuttal period.
>
> In the table below, we present for additional BEIR datasets the QPS at which LEMUR maintains the same NDCG@10 as exact MaxSim search. We will include these results in a further revision of the paper and we are happy to provide more detailed QPS-NDCG tradeoff results.
>
> | Dataset  | NDCG@10 | QPS  |
> |----------|---------|------|
> | SCIDOCS  | 15.8    | 1899 |
> | Quora    | 85.7    | 1119 |
> | NQ       | 56.1    | 181  |
> | HotPotQA | 67.8    | 62   |

---

> > ### Author Rebuttal · Reviewer_J12w · 2026-04-02
> >
> > Thanks to the authors for their response. Although several experiments are still missing, I find the paper interesting overall and will increase my score. I offer the following suggestions for the revised version:
> >
> > - **Include TREC DL 2019/2020 evaluations.**
> >   This is relatively straightforward once models are trained on MS MARCO. TREC DL 2019/2020 use the same document collection as MS MARCO (e.g., MS MARCO Dev), with the main difference being the query sets. While TREC DL contains fewer than 100 queries per year, it provides denser and higher-quality relevance annotations, which better reflect system performance. For implementation details, you may refer to the *Scaling Retriever* paper (https://arxiv.org/abs/2502.15526) and its codebase (https://github.com/HansiZeng/scaling-retriever).
> >
> > - **Expand BEIR evaluation.**
> >   Please include a more comprehensive evaluation on BEIR (e.g., all 13 or 15 datasets) and compare against strong baselines such as ColBERT-v2 in the refined paper. This would provide a clearer picture of the generalization ability of the proposed method.

---

> > > ### Author Response · Authors · 2026-04-04
> > >
> > > We thank the Reviewer for their helpful feedback and for updating their score. We chose to use the same datasets as earlier work for consistency (in particular, we use the same 6 datasets used in the MUVERA paper), but we are happy to perform the experiments also on the suggested datasets and will include them in the final revision of the paper.

---

### Decision · Program_Chairs · 2026-04-30

**Decision:**

Accept (regular)

**Comment:**

This paper proposes LEMUR, a method that approximates the late-interaction scoring function used in ColBERT with a learned single-vector representation. The idea of approximating a multi-vector late-interaction retriever with a learned single-vector representation is interesting. The paper provides a notable improvement on prior common approaches. The formulation of turning MaxSim into a multi-output regression problem is unique could be useful in other retrieval tasks as well. Reviewers have raised questions on difference from learning a dense retriever that distills ColBERT scores, evaluation using information retrieval metrics, comparison to retrieving-and-reranking using a strong single-vector retriever, cost of training the corpus-specific learned reduction, and robustness of LEMUR. The questions are mostly resolved, although some less-critical issues are only partially addressed. The contributions are acceptable as pointed by the reviewers. The authors are advised to add the related materials in the revised final version.